# WHICH HEADS MATTER FOR REASONING?
# RL-GUIDED KV CACHE COMPRESSION

## ABSTRACT

Reasoning large language models exhibit complex reasoning behaviors through the extended chain-of-thought generation, creating unprecedented Key-Value (KV) cache overhead during the decoding phase. Existing KV cache compression methods underperform on reasoning models: token-dropping methods break reasoning integrity by discarding critical information, while head-reallocation methods mistakenly compress reasoning-critical heads since they are designed for retrieval tasks, resulting in significant performance degradation as compression rates increase. We hypothesize that KV heads exhibit functional heterogeneity in reasoning models-some heads are critical for chain-of-thought consistency while others are compressible. To validate and exploit this insight, we propose RLKV, a novel reasoning-critical head identification method, which uses reinforcement learning to directly optimize the relationship between each head's cache usage and reasoning quality. As RLKV produces rewards from actual generated samples during training, it naturally identifies heads relevant to reasoning behaviors. We then allocate full KV cache to these heads while applying compressed constant KV cache to others for efficient inference. Our experiments reveal that only a small fraction of attention heads is essential for reasoning, enabling our KV compression approach to outperform baseline methods while achieving **20-50%** cache reduction with near lossless performance compared to uncompressed results.

## 1  INTRODUCTION

Recent advanced reasoning large language models (LLMs) (Jaech et al., 2024; Team et al., 2025; Guo et al., 2025; DeepMind, 2025) exhibit complex reasoning behaviors, such as self-reflection to revisit previous steps and exploration of alternative approaches, and achieve revolutionary performance on challenging mathematical and coding problems. However, this breakthrough creates an unprecedented memory bottleneck: the extension of chain-of-thought (CoT) reasoning generates significantly more tokens compared to conventional instruct models. For instance, Llama-3.1-8B-R1 (BF16) requires 16GB additional GPU memory for 32k CoT generation with a single query, primarily due to quadratic attention computation and linearly expanding KV cache. This limits large batch    FIX
processing and challenges the practical deployment of reasoning models.

Key-Value (KV) cache compression methods have demonstrated effectiveness for instruct models in long-context scenarios. As illustrated in Figure 1 (a), these methods typically follow one of two strategies: token dropping or head reallocation. Token-dropping methods selectively evict less important tokens from each head's KV cache (Zhang et al., 2023; Li et al., 2024; Cai et al., 2025; Yang et al., 2024b; Qin et al., 2024), while head-reallocation methods identify critical heads and allocate full KV cache to them, applying compressed KV cache to the remaining heads. However, as shown    FIX
in Figure 1 (b, left), two representative KV compression methods, including token-dropping method R-KV (Cai et al., 2025) and head-reallocation method DuoAttention (Xiao et al., 2024), degrade significantly when applied to reasoning models, while maintaining stable performance on their instruct counterparts. This performance degradation correlates strongly with generation length: in the MBPP (Austin et al., 2021) coding task, both model variants achieve nearly identical uncompressed performance, yet the reasoning variant generates on average 3341 tokens (approximately $8\times$) longer than the 439 tokens of the instruct variant. This controlled comparison isolates extended CoT generation as the primary cause of compression challenges, rather than differences in model capability, revealing the inherent difficulty of compressing long reasoning sequences. In reasoning models, the KV

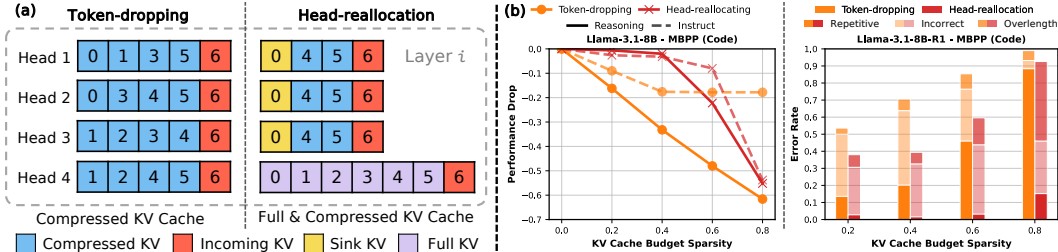

Figure 1: **(a) Overviews of Two Methods** *Left:* Token-dropping method removes less important tokens from each head's KV cache. *Right:* Head-reallocation method allocates full KV cache to critical heads while assigning constant-size KV cache to the remaining heads. **(b) Case study.** *Left:* The token-dropping method (R-KV) and the head-reallocation method (DuoAttention) maintain relatively stable performance on Llama-3.1-8B-Inst but degrade substantially on Llama-3.1-8B-R1, largely due to the longer generations produced by the reasoning model. *Right:* In terms of error modes, the token-dropping method (R-KV) tends to degenerate into repetitive behavior whereas the head-reallocation method (DuoAttention) often produces over-extended CoT that exhausts the length budget without reaching a correct solution. See Appendix A.1 for complete results.

cache undergoes a fundamental role shift: instead of serving merely as a computational optimization, it becomes the carrier of reasoning behaviors itself, storing critical states for CoT consistency and self-reflection, making compression inherently detrimental to reasoning performance.

To understand how the two KV-cache compression approaches underperform in preserving reasoning behaviors, we analyze their error modes as compression rates increase, as illustrated in Figure 1 (b, right). Models with token-dropping compression (R-KV) tend to lose reasoning behaviors because they inevitably discard reasoning-critical information, disrupting CoT consistency and leading to loops with repeated tokens. Although the R-KV approach (Cai et al., 2025) is designed specifically for reasoning models, it still cannot escape this inherent limitation. In contrast, models with head-reallocation compression (DuoAttention) relatively maintain coherent reasoning behaviors but are no longer effective: for problems that the uncompressed model can solve, the compressed model goes astray in its reasoning process and is unable to reach a solution within the maximum budget. This reveals that head-reallocation methods relatively preserve sequence information integrity in some heads by allocating full KV cache for them while compressing others (Xiao et al., 2023). However, they may mistakenly compress heads critical for reasoning behaviors, since their head identification targets "retrieval heads" (Wu et al., 2024). These methods rely on static patterns from prefill attention (Fu et al., 2024; Tang et al., 2024a) or single-forward-pass training (Xiao et al., 2024; Bhaskar et al., 2025), inherently failing to capture dynamic reasoning behaviors that emerge during extended CoT sequences.

These findings motivate our key insight that KV heads exhibit functional heterogeneity in reasoning models, where a subset of heads are critical to reasoning behaviors and naturally require a full KV cache to maintain CoT consistency. We term such heads with this role as **reasoning heads**. To validate and exploit this insight, we propose RLKV, a novel reasoning-critical head identification method, which employs reinforcement learning (RL) to identify those heads by directly optimizing the relationship between the allocation of each head's KV cache usage and reasoning quality. As illustrated in Figure 2, our method observes reasoning behaviors in generated samples and assigns rewards during RL training. These reward signals guide RL with sparsity pressure to optimize learnable gating adapters that control the mixing of full attention and local attention (Xiao et al., 2023). The gating adapters quantify each head's reliance on full versus local KV cache access, with L1 penalty encouraging sparsity. Through this RL optimization, the adapter values inherently distinguish *reasoning heads* from compressible heads, directly identifying which heads are essential for reasoning behaviors. In this way, our method consequently identifies *reasoning heads* and allocates full KV cache to them while applying compressed constant KV cache to others, effectively preserving reasoning behaviors during KV cache compression.

Our work makes three main contributions. First, we introduce RLKV, a novel reasoning-critical head identification method for guiding KV cache compression tailored to reasoning models, which leverages reward signals from RL training under sparsity pressure to directly supervise reasoning behaviors. Second, we achieve state-of-the-art compression performance, enabling near lossless reasoning capability with 20-50% KV cache usage reduction across diverse reasoning tasks and

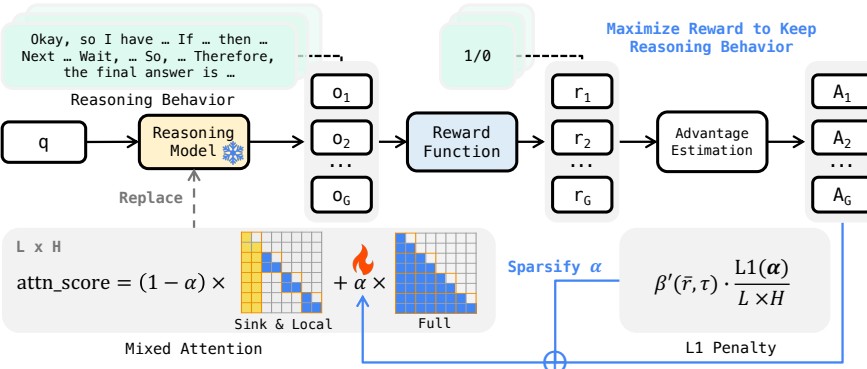

Figure 2: **Overview of RLKV:** Our method proposes to utilize RL to identify reasoning heads. The RL pipeline naturally captures reasoning behaviors, since it samples the current model's generations to produce reward signals. The reward function evaluates the samples to assess reasoning quality. We employ $L \times H$ learnable gating adapters to mix full attention and local attention for each head, quantifying each head's reliance on full versus local KV cache access. We apply an L1 penalty to encourage adapter sparsity, while RL optimizes the adapters to preserve reasoning behaviors. After training, we identify reasoning heads with high adapter values and allocate full KV cache to them while applying compressed KV cache to others for efficient inference.

models. Third, to our knowledge, RLKV is the first to identify a set of heads that matter for reasoning behaviors, while showing that other heads can still function under a compressed KV cache.           FIX

## 2 METHODOLOGY

In this section, we present RLKV, a novel reasoning-critical head identification method to guide efficient KV cache compression for reasoning LLMs, as illustrated in Figure 2. In this paper, we operationally define "***reasoning heads***" as the KV heads that:

*significantly degrade reasoning performance under local KV cache access.*

These identified *reasoning heads* are essential for reasoning behaviors, which naturally requires a full KV cache to maintain CoT consistency, while others are compressible. To achieve this, we first use mixed attention with gating adapters to quantify each head's reliance on complete or compressed KV cache usage. Then we apply RL with sparsity pressure to optimize the gating adapters based on a verifiable reward signal, naturally capturing reasoning behaviors. Finally, we introduce two complementary stabilization techniques to address the conflict between dense regularization and sparse rewards as the sparsity of adapters increases.

### 2.1 MIXED ATTENTION WITH GATING ADAPTERS

Identifying *reasoning heads* requires estimating individual KV heads' robustness of complete KV cache usage; therefore, we build upon mixed attention (Xiao et al., 2024), which uses lightweight gating adapters to quantify each head's reliance on full versus local KV cache access. Specifically, it combines two attention modes by attention mask, including full attention mapping to the full KV cache, and streaming attention (Xiao et al., 2023) mapping to the constant KV cache size containing initial sink tokens and recent tokens.

The mixed attention on each head can be formulated as:

$$\text{out\_mix\_attn}_{i,j} = \alpha_{i,j} \cdot \text{out\_full\_attn} + (1 - \alpha_{i,j}) \cdot \text{out\_streaming\_attn}, \quad (1)$$

where $\boldsymbol{\alpha} \in [0,1]^{L \times H}$ represents the learnable gating parameters for $L$ layers and $H$ heads, with $\alpha_{i,j}$ represents the weight assigned to full attention on the $j$-th head in the $i$-th layer. This design dramatically reduces the optimization space to only $L \times H$ gating parameters by freezing all LLM parameters, making it feasible to apply RL for identifying *reasoning heads*.

### 2.2 RL FOR REASONING HEAD IDENTIFICATION

Reasoning LLMs are often post-trained using reinforcement learning with verifiable reward (RLVR) (Guo et al., 2025; Team et al., 2025), which enhances reasoning capabilities by evaluating generated samples based solely on final answer correctness. During this RL training process, reasoning behaviors are naturally exhibited in the sampled CoT sequences, while reward signals directly reflect reasoning quality. These two characteristics make RLVR ideal for *reasoning heads* identification.

In concrete, we optimize the gating adapters $\boldsymbol{\alpha}$ using Group Relative Policy Optimization (GRPO) (Shao et al., 2024) on mathematical reasoning problems with two key modifications. First, to maximize the discriminative power of reward signals for *reasoning head* identification, we remove the KL penalty that conventionally limits reward signal strength to prevent over-optimization. Second, we apply L1 regularization (Tibshirani, 1996) to the adapters by incorporating the scaled L1 penalty term $\beta\|\boldsymbol{\alpha}\|_1/(L \times H)$ into the objective function to encourage adapter sparsity. The reward signal preserves high $\alpha_{i,j}$ values for *reasoning heads* requiring full KV cache access, while the L1 penalty drives $\alpha_{i,j}$ toward 0 for compressible heads.

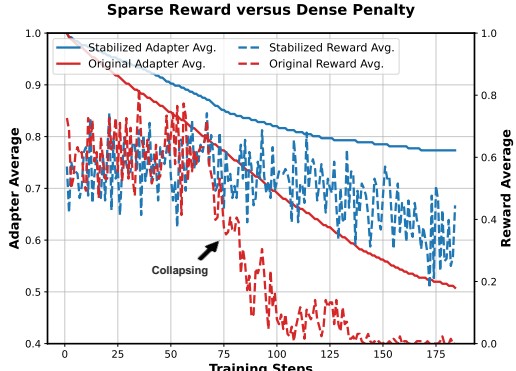

Figure 3: Gating adapter distribution after RLKV training on two models, which both are GQA architecture.

The overall objective is defined to maximize:

$$\underbrace{\frac{1}{G}\sum_{i=1}^{G}\min\left(\frac{\pi_{\boldsymbol{\alpha}}(o_i|q)}{\pi_{\boldsymbol{\alpha}_{\text{old}}}(o_i|q)}A_i, \text{clip}\left(\frac{\pi_{\boldsymbol{\alpha}}(o_i|q)}{\pi_{\boldsymbol{\alpha}_{\text{old}}}(o_i|q)}, 1-\epsilon, 1+\epsilon\right)A_i\right)}_{\text{reward signal}} - \underbrace{\frac{\beta}{L \times H}\|\boldsymbol{\alpha}\|_1}_{\text{L1 penalty}}, \quad (2)$$

where $q$ is the input query, $\{o_i\}_{i=1}^{G}$ are sampled outputs, $A_i$ is the normalized advantage, computed using a group of rewards $\{r_1, r_2, \cdots, r_G\}$ tailored to outputs:

$$A_i = \frac{r_i - \text{mean}(r_1, r_2, \cdots, r_G)}{\text{std}(r_1, r_2, \cdots, r_G)}. \quad (3)$$

The clipping mechanism with threshold $\epsilon$ prevents excessive policy updates, and $\beta$ controls the regularization strength. The policy $\pi_{\boldsymbol{\alpha}}$ represents the model's generation probability distribution conditioned on the current gating parameters $\boldsymbol{\alpha}$, and the advantage $A_i$ is positive for outputs leading to correct reasoning and negative for incorrect reasoning. This optimization naturally converges to a sparse solution where *reasoning heads* maintain high $\alpha$ values, as demonstrated in Figure 3

## 2.3 STABILIZATION FOR RL TRAINING

As adapters become increasingly sparse, the mixed attention of *reasoning heads* degenerates to the streaming attention, severely degrading the model's reasoning capacity, as shown in Figure 4. This degradation renders the reward signal increasingly sparse and unstable, while the L1 penalty remains dense across all parameters. This imbalance creates a vicious cycle, where degraded performance leads to sparser rewards, making the dense L1 penalty relatively stronger, which further drives adapters toward zero with no recovery capability. To resolve this destructive training dynamic and stabilize the training process, we introduce two complementary techniques that address this challenge from both the reward and penalty perspectives.

**Self-distillation Sampling.** Overly challenging problems during RL training lead to frequent failures and unstable reward signals. In

Figure 4: The conflict of sparse reward versus dense penalty leads to training collapse without our stabilization techniques. As adapters become sparse (decreasing average), model performance degrades (dropping reward), creating a vicious cycle where dense L1 penalties dominate increasingly sparse rewards.

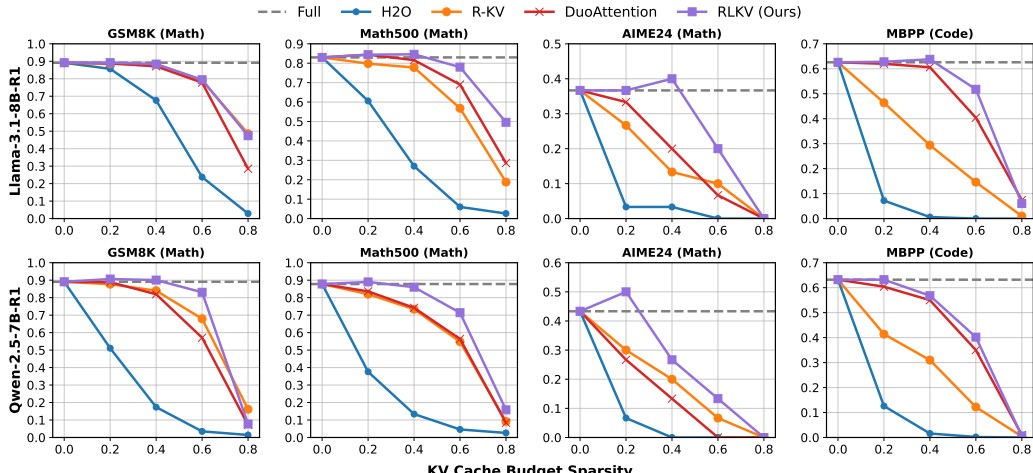

Figure 5: Performance comparison of RLKV against KV cache compression baselines across reasoning benchmarks. We evaluate RLKV (**Ours**) and existing methods on two reasoning models (Llama-3.1-8B-R1 and Qwen-2.5-7B-R1) across four benchmarks (GSM8K, MATH, AIME24, MBPP) at sparsity levels of 0.2, 0.4, 0.6, and 0.8. RLKV consistently outperforms all baselines across different sparsity levels, demonstrating particularly strong advantages at high sparsity levels (0.4 or 0.6) where competing methods suffer significant performance degradation. Complete numerical results are provided in Appendix A.3.

contrast to typical RLVR that utilizes sparse rewards for capability enhancement, our work leverages RL for capability preservation under sparsity constraints. Consequently, we focus on constructing high-quality training data that produces stable reward signals to improve learning efficiency. We construct training data by first filtering all problems the model initially solves correctly, then curating them to 3k using a curriculum sampling strategy (Team et al., 2025). We use output token lengths as a proxy for difficulty, enabling curriculum control that maintains stable reward signals throughout the training process. See Section 3.1 for training dataset details.

**Adaptive Penalty Weighting.** To address the penalty imbalance, we modulate the scaling weight $\beta$ of the L1 penalty based on the reward signal. Our design incorporates two protective mechanisms to prevent training collapse. First, we use adaptive scaling centered around a target reward of $\bar{r} \approx 0.7$ to smoothly decay penalty when performance degrades and increase it when performance improves. Second, we implement a hard cutoff at threshold $\tau$ to completely eliminate regularization when reasoning capability severely degrades. We implement this through a dynamic weight that replaces the constant hyperparameter $\beta$:

$$\beta'(\bar{r}, \tau) = \mathbb{I}(\bar{r} > \tau) \cdot \beta \cdot (\exp(\bar{r}) - 1), \quad \bar{r} = \text{mean}(r_1, r_2, \cdots, r_G), \tag{4}$$

where the exponential function $(\exp(\bar{r})-1)$ provides the adaptive scaling, and the indicator function $\mathbb{I}(\bar{r} > \tau)$ provides the hard cutoff based on mean reward $\bar{r}$ in the current group.

The end result is a set of identified *reasoning heads* that require full KV cache access, while non-reasoning heads can utilize compressed KV cache access, achieving significant memory compression without sacrificing reasoning capability. During inference, we use the learned gating parameters to rank all KV heads and select the top-k heads with the highest $\alpha$ values to maintain full KV cache access according to the target compression ratio. The remaining heads still use full attention but with compressed KV cache, which retains only initial sink tokens and recent tokens. Refer to Section 3.1 for further details of deployment and inference.

## 3 EXPERIMENTS

### 3.1 SETUPS

**Models, Datasets, and Baselines.** We evaluate RLKV on two mainstream small reasoning models, including Llama-3.1-8B-R1 and Qwen-2.5-7B-R1 (Guo et al., 2025), both are supervised fine-tuned from respective base models on DeepSeekR1 distilled CoT data (Guo et al., 2025). We con-

duct experiments on four benchmarks, using three datasets of increasing difficulty mathematical reasoning, GSM8K (Cobbe et al., 2021) for elementary problems, Math500 (Lightman et al., 2023) for intermediate problems and AIME24 (MMA, 2024) for advanced problems, to evaluate performance across difficulty levels, and MBPP (Austin et al., 2021) for Python programming to assess generalization beyond the training domain. We compare our method with KV cache compression FIX approaches including H$_2$O (Zhang et al., 2023) and R-KV (Cai et al., 2025), which are typical token-dropping methods, and DuoAttention (Xiao et al., 2024), which is a head-reallocation method.

**Implementation Details.** We implement RLKV by integrating MixedAttention into AReaL (Fu et al., 2025) and SGLang (Zheng et al., 2024). AReaL is an asynchronous distributed RL framework for updating adapters, and AReaL uses SGLang as the generation backend. We optimize gating adapters using GRPO with 4 samples per query and AdamW (Loshchilov & Hutter, 2017) with learning rate 0.01. We filter 3,000 mathematical problems from DeepScaleR (Luo et al., 2025) following our curriculum sampling strategy. During training, local attention uses 128 sink and 256 local tokens; for evaluation, non-reasoning heads use compressed KV cache only with 16 sink and 64 local tokens. To ensure fair comparison, we augment all baselines with equivalent token overhead and convert fixed-budget methods to dynamic allocation. Details are provided in Appendix A.2.

### 3.2 Main Results

Figure 5 presents the performance of RLKV against baselines across two reasoning models and four benchmarks at sparsity levels of 0.2, 0.4, 0.6, and 0.8. RLKV consistently outperforms all baselines at different levels of sparsity, with particularly strong advantages at high sparsity, such as 0.4 and 0.6, where other methods suffer significant performance degradation. Remarkably, RLKV even surpasses the full KV cache baseline on AIME24, the most challenging mathematical reasoning benchmark, for Llama-3.1-8B-R1 at 0.4 and Qwen-2.5-7B-R1 at 0.2, respectively. This counter-intuitive result suggests that our identified *reasoning heads* capture the essential components for complex reasoning, while non-reasoning heads may introduce noise that degrades performance when given full KV cache access. Notably, the performance degradation pattern at 0.8 sparsity directly reflects the relationship between *reasoning head* quantity and capability: as sparsity increases (retaining fewer reasoning heads), performance systematically decreases. This trend demonstrates that complex reasoning fundamentally depends on a sufficient number of *reasoning heads* with full KV cache access, making lossless compression at extreme ratios inherently challenging.

### 3.3 Analyses on Reasoning Heads versus Retrieval Heads

**Head Importance Analyses** Figure 6 presents head importance analyses by applying compressed KV cache to different types of heads: *reasoning heads* identified by RLKV, retrieval heads from DuoAttention, and randomly selected heads. We progressively replace the top fraction of heads with compressed KV cache and evaluate performance degradation on the Math500 benchmark. *Reasoning heads* identified by RLKV demonstrate significantly steeper performance degradation, indicating they are substantially more important than retrieval heads and random heads. Combined with the main results in

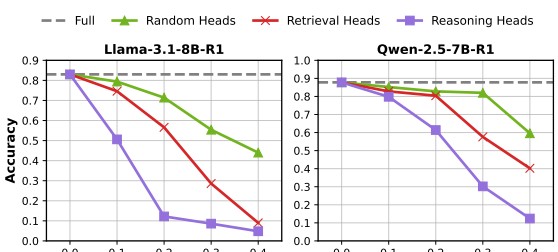

Figure 6: The importance of heads identified is equivalently illustrated by replacing the top ratio of them with a compressed KV cache. Compared to retrieval heads and random heads, reasoning heads identified by RLKV are more crucial to model performance, and are sensitive to compressed KV cache access.

Figure 5, this reveals an important asymmetry: compressing even a small fraction of top *reasoning heads* causes significant degradation, while maintaining complete capability requires preserving multiple *reasoning heads*. Qwen-2.5-7B-R1 shows more gradual degradation than Llama-3.1-8B-R1 at low compression ratios (0.1 and 0.2), indicating that its reasoning capability may be more distributed across multiple heads rather than concentrated in a few critical ones at these levels. Since Qwen-2.5-7B-R1 achieves stronger reasoning with fewer total heads (112 vs 256), it likely utilizes its top *reasoning heads* more efficiently, making it more robust to small-scale compression.

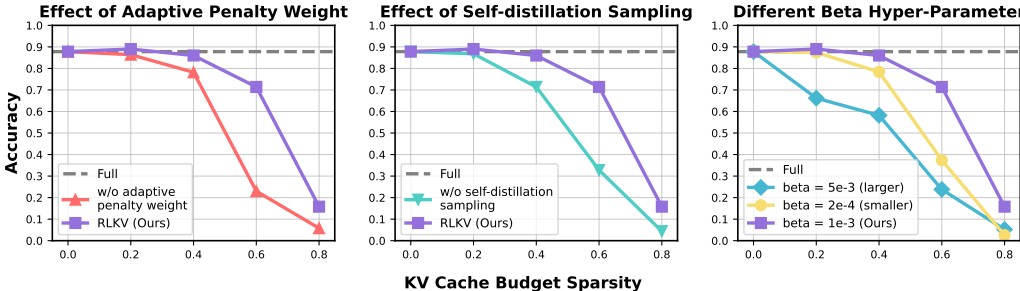

Figure 8: **Ablation study on key components of RLKV training method.** We evaluate three critical components using Qwen-2.5-7B-R1 on Math500. *Left*: Adaptive penalty weighting prevents training collapse by stabilizing conflicting dynamics between sparse rewards and L1 penalty, while its absence leads to ineffective exploration and training failure. *Middle*: Self-distillation sampling maintains stable reward signals by training on appropriately challenging problems, compared to unstable signals from overly difficult problems. *Right*: Base L1 penalty weight $\beta = 0.001$ achieves optimal sparsity-performance balance, while excessive penalty causes over-compression and insufficient penalty leads to premature convergence.

**Error Mode Analyses** We analyze the distinct error modes exhibited by models when *reasoning heads* and retrieval heads guide KV cache compression on the Math500 benchmark. Error modes are categorized into three types: repetitive errors (excessively repeating token sequences), incorrect errors (generating wrong answers), and overlength errors (generating sequences that exceed normal length baselines). Figure 7 reveals that models tend to produce repetitive generation errors when *reasoning heads* are compressed at higher levels, while models with compressed re-

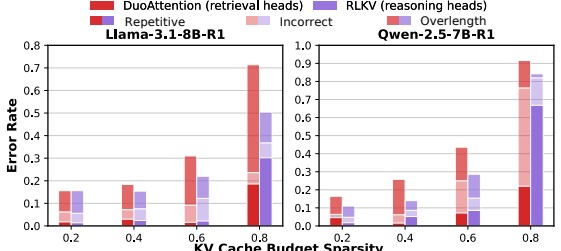

Figure 7: The analysis reveals distinct error modes when reasoning heads versus retrieval heads work with compressed KV cache on Math500 benchmark. Reasoning heads tend toward repetitive generation errors as compression increases, while retrieval heads exhibit more varied error modes across different settings.

trieval heads exhibit more varied error modes across different settings. This consistency in *reasoning head*-related errors suggests their collaborative role in maintaining complex logical states during reasoning, whereas retrieval heads appear to have more multifaceted roles. See Appendix A.4 for more details.

### 3.4 Memory Efficiency

To demonstrate RLKV's memory efficiency, we evaluate its compression performance while maintaining accuracy across two reasoning models and four benchmarks, as shown in Table 1 (a) and Table 1 (b). Values show performance with difference from full KV cache in parentheses, where light green indicates performance exceeding the full KV cache baseline and light red indicates performance below it. RLKV consistently outperforms baselines across all sparsity levels, achieving GPU memory reductions of 20-50% with minimal performance degradation across different models and benchmarks. Notably, different reasoning tasks exhibit varying sensitivity to compression, reflecting the heterogeneous and complex mechanisms underlying *reasoning head* functionality. When generation length exceeds 8k, 16k, or even 32k tokens, RLKV enables deployment on memory-constrained hardware and allows for higher inference parallelism by reducing memory bottlenecks.

### 3.5 Ablation Studies

We conduct ablation studies using Qwen-2.5-7B-R1 on the Math500 benchmark to assess the impact of adaptive penalty weighting, self-distillation sampling, and base L1 penalty weight in RLKV.

**Adaptive Penalty Weighting.** Figure 8 (left) demonstrates that adaptive penalty weighting significantly enhances performance by breaking the vicious cycle between sparse rewards and dense

Table 1: RLKV achieves near lossless performance (full KV cache) up to the sparsity thresholds shown for Llama-3.1-8B-R1 (a) and Qwen-2.5-7B-R1 (b) across four benchmarks. Red background denotes performance below the full–KV-cache baseline, whereas green background denotes performance above it. RLKV exhibits the smallest performance degradation among the other methods and, on some benchmarks, even improves over the full–KV-cache baseline. For all values, higher is better. The best result of the metric in each benchmark is in **bold**. All values are reported as percentages.

| Method | Lossless Sparsity Threshold | | | |
|---|---|---|---|---|
| | GSM8K (Math) 0.4 | Math500 (Math) 0.5 | AIME24 (Math) 0.4 | MBPP (Code) 0.4 |
| Full | 89.2 | 83.0 | 36.7 | 62.6 |
| H2O | 67.6 (-21.5) | 8.8 (-74.2) | 3.3 (-33.3) | 0.6 (-62) |
| R-KV | 88.3 (-0.8) | 68.6 (-14.4) | 13.3 (-23.3) | 29.4 (-33.2) |
| DuoAttention | 87.1 (-2.0) | 74.6 (-8.4) | 20 (-16.7) | 60.6 (-2) |
| RLKV (Ours) | **88.4** (-0.8) | **85** (+2) | **40** (+3.3) | **63.8** (+1.2) |

(a) Llama-3.1-8B-R1

| Method | Lossless Sparsity Threshold | | | |
|---|---|---|---|---|
| | GSM8K (Math) 0.4 | Math500 (Math) 0.4 | AIME24 (Math) 0.2 | MBPP (Code) 0.3 |
| Full | 89.1 | 87.8 | 43.3 | 63.2 |
| H2O | 17.4 (-71.7) | 13.4 (-74.4) | 6.7 (-36.7) | 2.2 (-61) |
| R-KV | 84 (-5.1) | 73.4 (-14.4) | 30 (-13.3) | 34.6 (-28.6) |
| DuoAttention | 82.0 (-7.1) | 74.2 (-13.6) | 26.7 (-16.7) | 59.4 (-3.8) |
| RLKV (Ours) | **90.1** (+1.0) | **86** (-1.8) | **50** (+6.7) | **62** (-1.2) |

(b) Qwen-2.5-7B-R1

L1 penalty. Without this mechanism, increasing adapter sparsity leads to degraded reasoning performance, which generates sparser reward signals while the L1 penalty remains dense, creating an imbalance that drives training toward collapse with no recovery capability.

**Self-distillation Sampling.** Self-distillation sampling provides stable reward signals throughout training, as shown in Figure 8 (middle). In contrast to typical RLVR that utilizes sparse rewards for capability enhancement, our work leverages RL for capability preservation under sparsity constraints. Training on problems suited to the model's reasoning capability maintains relatively stable reward signals throughout optimization, while training on overly challenging problems leads to unstable and sparse reward signals that provide weak and insufficient guidance for head identification.

**Base L1 penalty Weight.** The base regularization weight $\beta$ controls the strength of L1 penalty applied to gating adapters during RL training. Figure 8 (right) shows that a moderate $\beta$ value of 0.001 achieves an optimal balance between sparsity and reward signal strength. Excessive penalty ($\beta = 0.005$) dominates the optimization process, weakening reward signals through over-compression, while insufficient penalty ($\beta = 0.0002$) fails to induce adequate sparsity, leading to premature convergence with limited exploration of the reward landscape.

## 4 RELATED WORK

**Efficient LLM Inference.** Various techniques reduce KV cache overhead through architectural or system optimizations. Grouped-Query Attention (GQA) (Ainslie et al., 2023) and Multi-head Latent Attention (MLA) (Liu et al., 2024a) reduce the number of KV heads by sharing them across query heads, achieving significant memory reduction but requiring expensive pre-training from scratch. Linear attention methods (Gu & Dao, 2023; Yang et al., 2025b) maintain constant memory usage during inference by avoiding the quadratic attention computation, but exhibit reduced modeling capacity compared to standard transformer architectures. KV cache quantization (Liu et al., 2024b; Tao et al., 2025; Hooper et al., 2024; Duanmu et al., 2024; Su et al., 2025; Yue et al., 2024) and system-level optimizations, such as paged KV cache (Kwon et al., 2023), KV cache reuse (Zheng

et al., 2024), and sparsely loading KV cache (Tang et al., 2024b), provide orthogonal improvements by reducing the precision or optimizing the storage/retrieval of cached states. While valuable, these methods treat KV cache as opaque data without exploiting the inherent sparsity patterns.

**KV Cache Compression.**   Recent works mainly exploit sparsity in long-context scenarios for instruct models, including token-dropping and head-reallocation methods. (1) Token-dropping methods (Zhang et al., 2023; Li et al., 2024; Cai et al., 2025; Yang et al., 2024b; Qin et al., 2024) apply eviction strategies across all heads or intra-layer heads based on attention scores. $H_2O$ (Zhang et al., 2023) maintains important tokens' KV cache based on accumulated attention scores plus a sliding window for recent tokens. Specifically, recent R-KV (Cai et al., 2025), designed for reasoning models, primarily adds similarity-based clustering to priority evict redundancy tokens' KV cache during both prefill and decoding phases. However, they inevitably discard reasoning-critical information and disrupt the CoT consistency as compression rates increase. (2) head-reallocation methods (Fu et al., 2024; Tang et al., 2024a; Xiao et al., 2024; Bhaskar et al., 2025) maintain full KV cache only for identified retrieval heads (Wu et al., 2024) in long-context scenarios while applying compressed KV cache (Xiao et al., 2023) to others. Ada-KV (Fu et al., 2024) and RazorAttention (Tang et al., 2024a) use proxy metrics of attention scores, while DuoAttention (Xiao et al., 2024) and PruLong (Bhaskar et al., 2025) are learning-based methods for head identification. DuoAttention minimizes single-forward output deviation on a synthetic long-context recall task, while PruLong uses next-token loss on long-context pre-training corpora. However, these methods do not capture the reasoning behaviors that emerge during dynamically extending CoT generation, resulting in degraded reasoning performance as compression rates increase.

**Reinforcement Learning for Efficiency.**   RL has proven effective in Neural architecture search (Zoph & Le, 2017; Zoph et al., 2018), where it treats architecture choices as sequential decisions, and model pruning (He et al., 2018), where it learns layer-wise pruning ratios that maximize accuracy under resource constraints. However, the limitation is the high computational cost due to the large optimization space. Our work utilizes gating values assigned to each KV head to reduce the optimization space and make RL feasible and efficient. For reasoning language models, recent works apply RL tuning to mitigate overthinking (Hou et al., 2025; Liu et al., 2025) by learning to reduce CoT length while maintaining reasoning capability, thereby indirectly decreasing KV cache requirements. Our work is orthogonal to these methods, employing lightweight RL training to identify *reasoning heads* that guide KV cache compression while preserving reasoning capability.

## 5 CONCLUSION

In this paper, we propose RLKV, a novel reasoning-critical head identification method to guide KV cache compression in reasoning models. RLKV directly optimizes the relationship between each head's KV cache usage and reasoning quality through reinforcement learning and we achieve competitive performance at diverse KV cache budget sparsity levels and reduce 20-50% KV cache usage while preserving full reasoning capability across Llama-3.1-8B-R1 and Qwen-2.5-7B-R1 on GSM8K, MATH, AIME24, and MBPP benchmarks. Then we analyze the *reasoning heads* importance and error modes, revealing the importance and complexity of *reasoning heads* in reasoning models. RLKV provides a new perspective on understanding reasoning models and opens up new avenues for efficient inference of reasoning LLMs.

## 6 FUTURE WORK

RLKV opens several promising avenues for future research. First, the significant variability in *reasoning heads* distribution across different models and tasks presents an exciting opportunity to develop a deeper understanding of the heterogeneous nature of reasoning mechanisms in reasoning LLMs. Second, while RLKV effectively identifies *reasoning heads* for compression, exploring the complete functional roles of these heads beyond reasoning could unlock new insights into model interpretability and architectural design. Third, advancing compression techniques to maintain strong performance at extremely high compression ratios (80% and above) represents a compelling challenge that could further bridge the gap between memory efficiency and reasoning capability preservation. These research directions hold significant potential for advancing both our understanding of reasoning in large language models and their practical deployment efficiency.

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

## DECLARATION OF THE USE OF LARGE LANGUAGE MODELS

In this paper, we only use LLMs to help with grammar checking and polishing the writing. All conceptual contributions, framework design, implementation, and experimental evaluations were performed by the authors without assistance from LLMs.

## A APPENDIX

### A.1 MOTIVATION STUDY

We provide a comprehensive motivation study on two mainstream small reasoning models (Llama-3.1-8B-R1 and Qwen-2.5-7B-R1 (Guo et al., 2025)) and their instruct variants (Llama-3.1-8B-Inst (Dubey et al., 2024) and Qwen-2.5-7B-Inst [1] (Yang et al., 2024a)). We conduct the evaluation on two typical token-dropping methods ($H_2O$ (Zhang et al., 2023) and R-KV (Cai et al., 2025)) and one head-reallocation method (DuoAttention (Xiao et al., 2024)) across four benchmarks, including GSM8K (Cobbe et al., 2021), Math500 (Lightman et al., 2023), AIME24(MMA, 2024), MBPP (Austin et al., 2021). Figure 9 presents that all compression methods maintain relatively stable performance on instruct models but drop substantially on reasoning models as compression increases.                          FIX

We further analyze the error modes on reasoning models in the above evaluation. We observed three error modes: repetitive errors (excessively repeating token sequences), incorrect errors (generating wrong answers), and overlength errors (generating sequences that exceed normal length baselines), as illustrated in Figure 10. The detailed error modes can be seen in Figure 11.

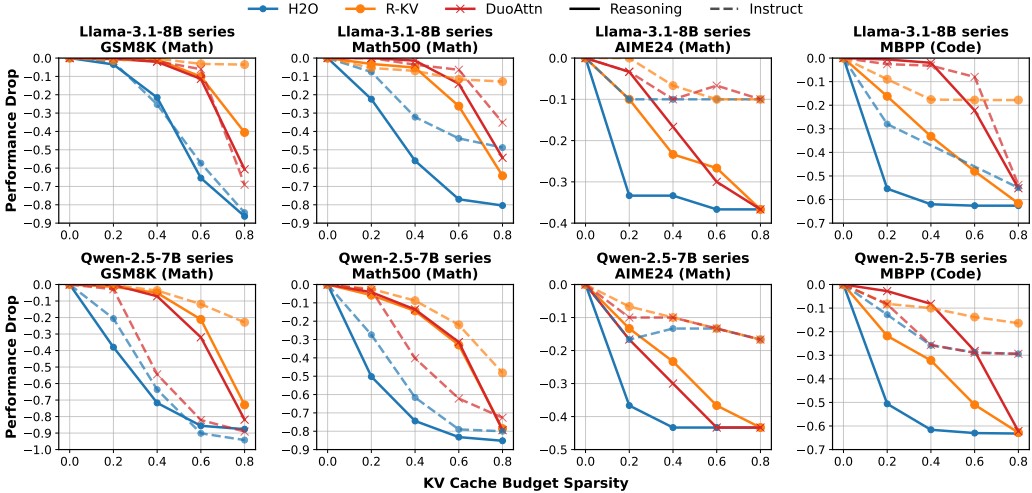

Figure 9: Comprehensive evaluation of KV cache compression methods across all model pairs and benchmarks reveals consistent patterns of performance degradation. $H_2O$, R-KV, and DuoAttention maintain relatively stable performance on instruction-following models but exhibit significant drops on their reasoning counterparts as the KV cache budget decreases. This performance degradation becomes particularly severe at higher sparsity levels, with notable declines observed on reasoning-intensive benchmarks including GSM8k, Math500, AIME24, and MBPP.                          FIX

### A.2 EXPERIMENT DETAILS

**Dataset Construction.** We construct training data from the DeepScaleR dataset (Luo et al., 2025), which contains about 40,000 diverse and challenging mathematical reasoning problems. For each model, we generate solutions using the respective reasoning model with greedy decoding, filter

---

[1] We use Qwen-2.5-Math-7B-Instruct (Yang et al., 2024a) as the instruct baseline, abbreviated as Qwen-2.5-7B-Inst for naming consistency, since Qwen-2.5-7B-R1 (deepseek-ai/DeepSeek-R1-Distill-Qwen-7B) was based on Qwen-2.5-Math-7B

**Repetitive Error:** Okay, so I have this problem here where I need to find the greatest four-digit number N such that if you change any of its digits to 1, the resulting number is divisible by 7. Then, I have to find Q + R where Q is the quotient and R is the remainder when N is divided by 1000. ...(after about 2000 tokens) but that's not necessarily the case because 1111 is 7*11*13, which is 7*143, but 1111 is 7*11*13, so 1111 is 7*11*13, ==which is 7*11*13==, ==which is 7*11*13==, ==which is 7*11*13==, ==which is 7*11*13== ...(keep repeating)

**Incorrect Error:** Okay, so I have this problem about residents in Aimeville and the things they own. Let me try to figure it out step by step. Hmm, it's a problem involving sets and maybe using some principles from set theory or combinatorics. I remember something about inclusion-exclusion principles from my math classes, so maybe that's what I need here. ... (after about 6500 tokens) Therefore, the number of residents who own all four things is \\(\\boxed{219}\\). #but correct answer is 73

**Overlength Error:** Okay, so I have this probability problem here: there are four points, A, B, C, and D, ... In this diagram, the green edges represent the labeling where $AB$ and $CD$ intersect, and the blue and red edges represent the equally likely labelings where $AB$ and $CD$ do not intersect #stop at 8k maximum output length

Figure 10: The instances of three error modes.

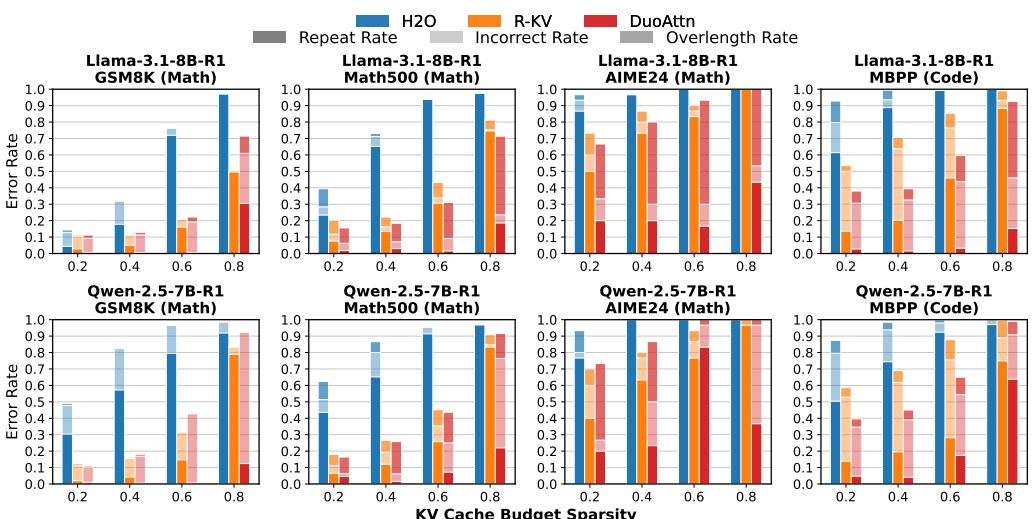

Figure 11: Comprehensive error mode analyses of KV cache compression methods across reasoning models reveal distinct failure patterns. Token-dropping methods (H$_2$O, R-KV) consistently exhibit repetitive errors, as they inevitably discard reasoning-critical information during compression. In contrast, the head-reallocation method DuoAttention tends to show more over-length errors compared to token-dropping methods, suggesting that while it relatively preserves sequence information integrity, it still struggles to fully preserve reasoning capability.

correct solutions, then randomly sample 3,000 problems for training. The selected problems are distributed across different output token lengths as follows: 600 problems each for 0-2k and 2k-4k tokens, 1,000 problems for 4k-6k tokens, and 800 problems for 6k-8k tokens.

**Hardware and Hyperparameter Settings.** All experiments are conducted on 2 NVIDIA A100 GPUs (80GB) for several hours, one for backward computation and one for sample generation. Training runs for 2 epochs, totaling 185 steps with a batch size of 32. All evaluations are conducted on NVIDIA RTX5090 GPUs. We optimize the gating adapters using AdamW optimizer with $\beta_1 = 0.9$, $\beta_2 = 0.999$, weight decay of 0.017, and learning rate of 0.01 with constant schedule. For GRPO training configuration, we disable KL penalty and use recommendation setting of AReaL; for GRPO sampling configuration, we use 4 samples per query with sampling temperature of 1.0. The hyperparameters are shown in Table 2.

**Local Attention Implementation.** During training, we employ an efficient block-sparse attention approximation implementation (Guo et al., 2024) in AReaL (Fu et al., 2025) to update adapter weights, while using mask matrices for prefilling and custom Triton kernels for decoding in SGLang (Zheng et al., 2024) to generate samples. For inference, we only store the partial KV cache of first 16 sink tokens and recent 64 local tokens for non-reasoning heads, while *reasoning heads* maintain the full KV cache.

Table 2: Training Hyperparameters.

| Parameter | Llama-3.1-8B-R1 | Qwen-2.5-7B-R1 |
|---|---|---|
| Regularization weight $\beta$ | 1e-3 | 1e-3 |
| Reward threshold $\tau$ | 0.5 | 0.55 |
| Sink token size | 128 | 128 |
| Local token size | 256 | 256 |
| Max sequence length | 8192 | 8192 |

**Baseline Implementation.** To ensure fair comparison with baseline methods, we make several adjustments. For $H_2O$ and R-KV, we augment them with the same sink and local token overhead (16+64 tokens) that our method uses. Since $H_2O$ and R-KV only support preset fixed KV cache budgets, we convert their fixed budgets to dynamic allocation that increases with sequence length. For example, if the fixed budget is 50% of the full KV cache, then at sequence length 1000, they use 500 tokens of KV cache, and at sequence length 2000, they use 1000 tokens of KV cache. For DuoAttention, we replicate their approach with default settings on our models and use the same inference settings as our method.

**Training Cost.** The training of the adapters is computationally modest: on 2 A100 GPUs, our method consumes 40, 22, and 36 GPU-hours for Llama-3.1-8B-R1, Qwen-2.5-7B-R1, and Qwen-3-4B-Thinking, respectively. FIX

**Evaluation Settings.** We evaluate all methods using greedy decoding on RTX 5090 36G GPUs or RTX 4090 24G GPUs with batch size of 1. For all datasets, we use regex to extract the final answer from the generated text, using Pass@1 as the evaluation metric. For GSM8K, Math500, and MBPP, we use 8192 max sequence length; for AIME24, we use 16384 max sequence length. We achieved near official reported performance without KV cache compression. We use eager attention implementation for $H_2O$ and R-KV since they need to use attention scores, while we use flash attention for DuoAttention and our method.

**Prompt Template.** We follow the prompt setting recommended by DeepSeek-R1 (Guo et al., FIX 2025) in both training and evaluation without additional prompt engineering. For example, we use the following template in math problems:

```
Solve the following math problem efficiently and
clearly.  The last line of your response should
be of the following format:  'Therefore, the final
answer is:  $\\boxed{ANSWER}$.  I hope it is correct'
(without quotes) where ANSWER is just the final number
or expression that solves the problem.  Think step by
step before answering.

QUESTION
```

## A.3 FULL RESULTS

Tables 3 and 4 present the complete numerical results of RLKV and baselines for Llama-3.1-8B-R1 and Qwen-2.5-7B-R1 respectively, across all benchmarks and KV cache compression budgets. Values in parentheses indicate the performance difference compared to the full KV cache setting, with positive values in green indicating improvement and negative values in red indicating degradation.

## A.4 DETAILS OF ERROR MODES ANALYSES

Figure 12 presents the comprehensive error mode analysis across all models and benchmarks. We observe three error modes: repetitive errors (excessively repeating token sequences), incorrect errors (generating wrong answers), and overlength errors (generating sequences that exceed normal length baselines). Our method RLKV shows consistency in error modes across different models and benchmarks, while DuoAttention exhibits more varied error modes across different settings.

Table 3: Llama-3.1-8B-R1 performance (%) under different KV cache compression methods and budgets. RLKV (**Ours**) shows competitive performance across settings. Red background denotes performance below the full–KV-cache baseline, whereas green background denotes performance above it. For all values, higher is better. The best result of the metric in each benchmark is in **bold**.

| Dataset | Method | KV Cache Budget Sparsity | | | |
|---|---|---|---|---|---|
| | | 0.2 | 0.4 | 0.6 | 0.8 |
| GSM8K (Math) | H2O | 85.7 (-3.5) | 67.6 (-21.5) | 23.7 (-65.4) | 2.9 (-86.3) |
| | R-KV | 88.5 (-0.6) | 88.3 (-0.8) | 79.1 (-10.1) | **48.6** (-40.6) |
| | DuoAttention | 88.8 (-0.4) | 87.1 (-2.0) | 77.8 (-11.4) | 28.5 (-60.6) |
| | RLKV (Ours) | 89.2 (+0.1) | **88.4** (-0.8) | **79.5** (-9.7) | 47.4 (-41.8) |
| Math500 (Math) | H2O | 60.6 (-22.4) | 27.0 (-56.0) | 6.0 (-77.0) | 2.6 (-80.4) |
| | R-KV | 79.8 (-3.2) | 77.8 (-5.2) | 56.8 (-26.2) | 18.8 (-64.2) |
| | DuoAttention | 84.4 (+1.4) | 81.6 (-1.4) | 69.0 (-14.0) | 28.6 (-54.4) |
| | RLKV (Ours) | 84.4 (+1.4) | **84.6** (+1.6) | **78.0** (-5.0) | 49.6 (-33.4) |
| AIME24 (Math) | H2O | 3.3 (-33.3) | 3.3 (-33.3) | 0.0 (-36.7) | **0.0** (-36.7) |
| | R-KV | 26.7 (-10.0) | 13.3 (-23.3) | 10.0 (-26.7) | **0.0** (-36.7) |
| | DuoAttention | 33.3 (-3.3) | 20.0 (-16.7) | 6.7 (-30.0) | **0.0** (-36.7) |
| | RLKV (Ours) | 36.7 (+0.0) | **40.0** (+3.3) | **20.0** (-16.7) | **0.0** (-36.7) |
| MBPP (Code) | H2O | 7.2 (-55.4) | 0.6 (-62.0) | 0.0 (-62.6) | 0.0 (-62.6) |
| | R-KV | 46.4 (-16.2) | 29.4 (-33.2) | 14.6 (-48.0) | 1.0 (-61.6) |
| | DuoAttention | 62.0 (-0.6) | 60.6 (-2.0) | 40.4 (-22.2) | **7.4** (-55.2) |
| | RLKV (Ours) | **62.8** (+0.2) | **63.8** (+1.2) | **51.8** (-10.8) | 6.0 (-56.6) |

Table 4: Qwen-2.5-7B-R1 performance (%) under different KV cache compression methods and budgets. RLKV (**Ours**) shows competitive performance across settings. Red background denotes performance below the full–KV-cache baseline, whereas green background denotes performance above it. For all values, higher is better. The best result of the metric in each benchmark is in **bold**.

| Dataset | Method | KV Cache Budget Sparsity | | | |
|---|---|---|---|---|---|
| | | 0.2 | 0.4 | 0.6 | 0.8 |
| GSM8K (Math) | H2O | 51.1 (-38.0) | 17.4 (-71.7) | 3.5 (-85.6) | 1.4 (-87.6) |
| | R-KV | 87.7 (-1.4) | 84.0 (-5.1) | 67.9 (-21.1) | **16.1** (-72.9) |
| | DuoAttention | 88.9 (-0.2) | 82.0 (-7.1) | 57.1 (-32.0) | 7.2 (-81.9) |
| | RLKV (Ours) | **90.7** (+1.6) | **90.1** (+1.0) | **83.1** (-6.0) | 7.7 (-81.3) |
| Math500 (Math) | H2O | 37.6 (-50.2) | 13.4 (-74.4) | 4.6 (-83.2) | 2.6 (-85.2) |
| | R-KV | 82.0 (-5.8) | 73.4 (-14.4) | 54.8 (-33.0) | 9.2 (-78.6) |
| | DuoAttention | 83.6 (-4.2) | 74.2 (-13.6) | 56.4 (-31.4) | 8.4 (-79.4) |
| | RLKV (Ours) | **89.0** (+1.2) | **86.0** (-1.8) | **71.4** (-16.4) | **15.8** (-72.0) |
| AIME24 (Math) | H2O | 6.7 (-36.7) | 0.0 (-43.3) | 0.0 (-43.3) | **0.0** (-43.3) |
| | R-KV | 30.0 (-13.3) | 20.0 (-23.3) | 6.7 (-36.7) | **0.0** (-43.3) |
| | DuoAttention | 26.7 (-16.7) | 13.3 (-30.0) | 0.0 (-43.3) | **0.0** (-43.3) |
| | RLKV (Ours) | **50.0** (+6.7) | **26.7** (-16.7) | **13.3** (-30.0) | **0.0** (-43.3) |
| MBPP (Code) | H2O | 12.6 (-50.6) | 1.6 (-61.6) | 0.2 (-63.0) | 0.0 (-63.2) |
| | R-KV | 41.4 (-21.8) | 31.0 (-32.2) | 12.2 (-51.0) | 0.4 (-62.8) |
| | DuoAttention | 60.4 (-2.8) | 55.0 (-8.2) | 35.0 (-28.2) | **1.0** (-62.2) |
| | RLKV (Ours) | **63.2** (+0.0) | **56.8** (-6.4) | **40.2** (-23.0) | 0.6 (-62.6) |

## A.5 EVALUATION ON QWEN-3-4B-THINKING

NEW

We further evaluate RLKV and the baselines on Qwen-3-4B-Thinking, a newly released and powerful reasoning model (Yang et al., 2025a) to validate the effectiveness of our method. The evaluation is conducted on four reasoning benchmarks (GSM8K, MATH, AIME24, MBPP) at sparsity levels of 0.2, 0.4, 0.6, and 0.8, following the same settings as in the main experiment. As shown in Figure 13 and Table 5, RLKV on Qwen-3-4B-Thinking exhibits performance trends similar to those ob-

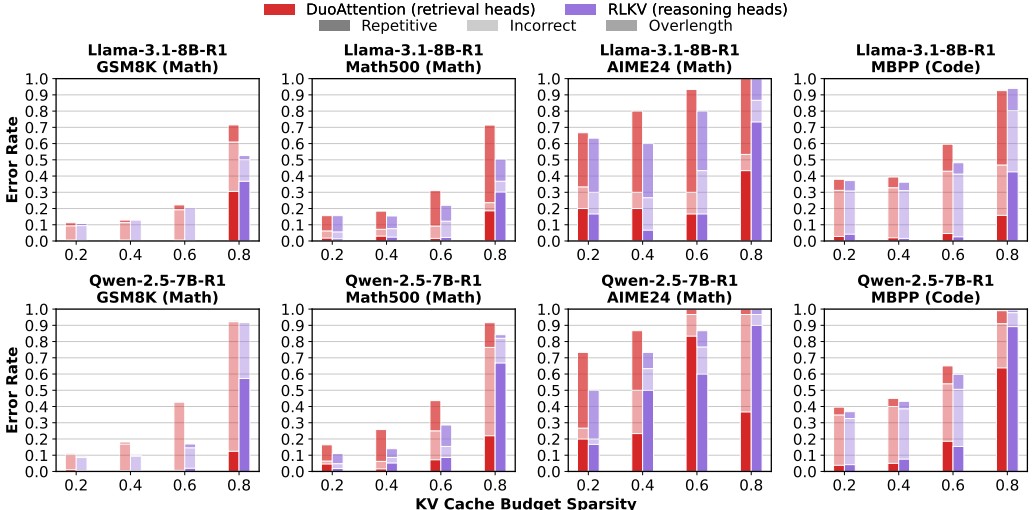

Figure 12: The analysis reveals distinct error patterns when reasoning heads versus retrieval heads work with compressed KV cache across four benchmarks.

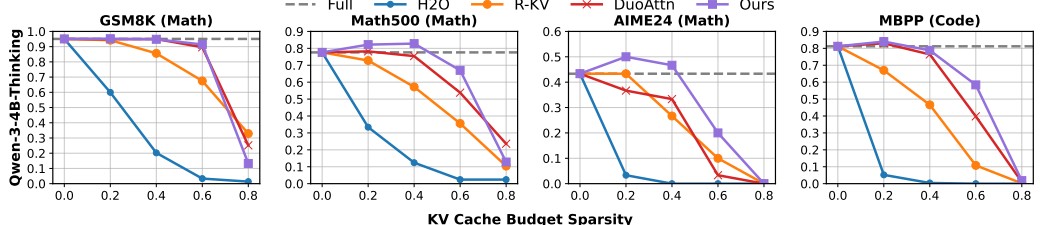

Figure 13: Performance comparison of RLKV against KV cache compression baselines across reasoning benchmarks. We evaluate RLKV (**Ours**) and existing methods on Qwen-3-4B-Thinking across four benchmarks (GSM8K, MATH, AIME24, MBPP) at sparsity levels of 0.2, 0.4, 0.6, and 0.8. RLKV consistently outperforms all baselines across 0.2-0.6 sparsity levels, but performance drops significantly at 0.8 sparsity due to extreme compression. The results demonstrate particularly strong advantages at 0.4 or 0.6 sparsity levels where competing methods suffer significant performance degradation.

served in our previous evaluations on Llama-3.1-8B-R1 and Qwen-2.5-7B-R1. RLKV outperforms all baselines at sparsity levels 0.2, 0.4, and 0.6, but, due to extreme compression, suffers performance drops at 0.8 sparsity, similar to the baselines. Table 6 shows the maximum sparsity at which RLKV on Qwen-3-4B-Thinking maintains lossless performance compared to the uncompressed settings across the four benchmarks. RLKV achieves 50% memory reduction on GSM8K, Math500, and AIME24, and 30% on MBPP, while the baselines suffer significant performance degradation at these sparsity levels. Compared to the results on Llama-3.1-8B-R1 and Qwen-2.5-7B-R1, RLKV on Qwen-3-4B-Thinking attains a higher maximum sparsity without performance loss, suggesting better compression capability on stronger models. This trend further supports the effectiveness of RLKV across different model architectures and scales.

## A.6 FOUR SUBSETS OF MMLU-PRO ON LLAMA-3.1-8B-R1 AND QWEN-2.5-7B-R1  NEW

We further validate RLKV on generalization beyond the training math domain. We evaluate RLKV and baselines on the four subsets of the challenging knowledge QA benchmark MMLU-Pro (Wang et al., 2024), including MMLU-Pro-Chemistry, MMLU-Pro-Computer-Science, MMLU-Pro-Law, and MMLU-Pro-Physics. Due to the time constraints, we randomly sample 200 examples from each subset for evaluation on Llama-3.1-8B-R1 and Qwen-2.5-7B-R1. As shown in Figure 14, RLKV achieves comparable or better accuracy than baselines on these four subsets across four sparsity settings (0.2, 0.4, 0.6, 0.8). For Law on Qwen-2.5-7B-R1 and Physics on Llama-3.1-8B-R1, RLKV

Table 5: Qwen-3-4B-Thinking performance (%) under different KV cache compression methods and budgets. RLKV (**Ours**) shows competitive performance across settings. Red background denotes performance below the full–KV-cache baseline, whereas green background denotes performance above it. For all values, higher is better. The best result of the metric in each benchmark is in **bold**.

| Dataset | Method | KV Cache Budget Sparsity | | | |
| --- | --- | --- | --- | --- | --- |
| | | 0.2 | 0.4 | 0.6 | 0.8 |
| GSM8K (Math) | RLKV | **95.2** (+0.2) | 94.8 (-0.2) | **91.4** (-3.6) | 13.3 (-81.8) |
| | DuoAttention | 94.8 (-0.2) | **95.0** (-0.1) | 89.6 (-5.5) | 25.2 (-69.9) |
| | R-KV | 94.2 (-0.8) | 85.6 (-9.5) | 67.5 (-27.6) | **32.9** (-62.2) |
| | H2O | 60.0 (-35.1) | 20.2 (-74.8) | 3.3 (-91.7) | 1.4 (-93.7) |
| Math500 (Math) | RLKV | **82.2** (+4.6) | **82.8** (+5.2) | **67.0** (-10.6) | 12.8 (-64.8) |
| | DuoAttention | 78.2 (+0.6) | 75.4 (-2.2) | 53.8 (-23.8) | **23.6** (-54.0) |
| | R-KV | 72.8 (-4.8) | 57.2 (-20.4) | 35.6 (-42.0) | 10.4 (-67.2) |
| | H2O | 33.4 (-44.2) | 12.4 (-65.2) | 2.4 (-75.2) | 2.4 (-75.2) |
| AIME24 (Math) | RLKV | **50.0** (+6.7) | **46.7** (+3.3) | **20.0** (-23.3) | **0.0** (-43.3) |
| | DuoAttention | 36.7 (-6.7) | 33.3 (-10.0) | 3.3 (-40.0) | **0.0** (-43.3) |
| | R-KV | 43.3 (+0.0) | 26.7 (-16.7) | 10.0 (-33.3) | **0.0** (-43.3) |
| | H2O | 3.3 (-40.0) | 0.0 (-43.3) | 0.0 (-43.3) | **0.0** (-43.3) |
| MBPP (Code) | RLKV | **84.0** (+2.8) | **78.8** (-2.4) | **58.4** (-22.8) | **1.8** (-79.4) |
| | DuoAttention | 83.0 (+1.8) | 76.4 (-4.8) | 39.8 (-41.4) | 1.0 (-80.2) |
| | R-KV | 67.0 (-14.2) | 46.6 (-34.6) | 10.8 (-70.4) | 0.2 (-81.0) |
| | H2O | 5.2 (-76.0) | 0.4 (-80.8) | 0.0 (-81.2) | 0.0 (-81.2) |

Table 6: RLKV achieves near lossless performance (full KV cache) up to the sparsity thresholds shown for Qwen-3-4B-Thinking across four benchmarks. Red background denotes performance below the full-KV-cache baseline, whereas green background denotes performance above it. RLKV exhibits the smallest performance degradation among the other methods and, on some benchmarks, even improves over the full-KV-cache baseline. For all values, higher is better. The best result of the metric in each benchmark is in **bold**. All values are reported as percentages.

| Method | Lossless KV Cache Budget Sparsity on each Dataset | | | |
| --- | --- | --- | --- | --- |
| | GSM8K (Math) 0.5 | Math500 (Math) 0.5 | AIME24 (Math) 0.5 | MBPP (Code) 0.3 |
| RLKV | **93.9** (-1.1) | **82.8** (+5.2) | **43.3** (+0) | **82.4** (+1.2) |
| DuoAttention | 93.0 (-2.0) | 72.4 (-5.2) | 20 (-23.3) | 80.2 (-1) |
| R-KV | 75.8 (-19.2) | 51.6 (-26.0) | 13.3 (-30) | 58 (-23.2) |
| H2O | 9.5 (-85.6) | 5.4 (-72.2) | 0 (-43.3) | 0.4 (-80.8) |

cannot achieve near lossless compression even at sparsity 0.2. Although this suggests a limitation in RLKV for these specific model-task combinations, it still outperforms other methods.

## A.7 AN IMPLICITLY UNFAIR COMPARISON IN FIXED-BUDGET EVALUATION

NEW

This section discusses the motivation for using a dynamic budget instead of a fixed budget for KV cache compression evaluation. Existing long-context compression works (Li et al., 2024; Yang et al., 2024b; Qin et al., 2024; Fu et al., 2024; Tang et al., 2024a; Xiao et al., 2024; Bhaskar et al., 2025) typically evaluate on in-context recall tasks, where each sample's prompt length is fixed/controlled. A fixed budget of the form budget = sparsity × prompt_length then yields a roughly consistent compression ratio per sample, so fixed budgets are fair in that setting.

For reasoning tasks, however, the response length is often much larger than the prompt, as shown in Figure 15. If we use a global fixed budget (e.g., 1k tokens), any sample whose full output fits

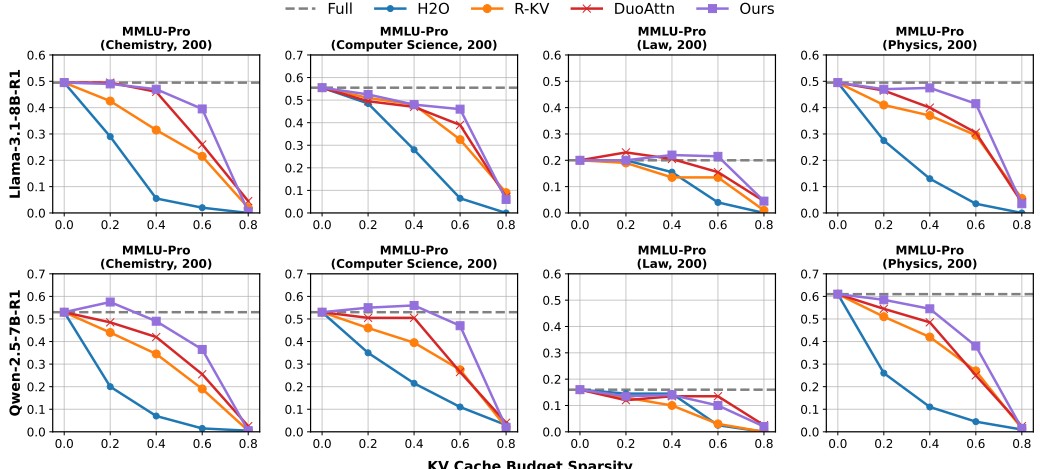

Figure 14: Performance comparison of RLKV against KV cache compression baselines across four subsets of the MMLU-Pro benchmark, including Chemistry, Computer Science, Law, and Physics. We evaluate RLKV (**Ours**) and existing methods on two reasoning models (Llama-3.1-8B-R1 and Qwen-2.5-7B-R1) across four benchmarks (GSM8K, MATH, AIME24, MBPP) at sparsity levels of 0.2, 0.4, 0.6, and 0.8. RLKV consistently outperforms all baselines across different sparsity levels, demonstrating the generalization beyond the reasoning domain.

within 1k tokens is uncompressed, while longer samples are compressed. Thus, different samples experience very different compression ratios, and fixed budgets are not fair at the per-sample level.

In R-KV (Cai et al., 2025), the reported compression rate is computed as budget/average_full_length. For example, R-KV achieves the compression ratio of 66.2% for Math500 on Llama-3.1-8B-R1, with a fixed budget of 200 and an average full length of 3019. However, a large fraction of samples are uncompressed and thus produce the same responses as the full model. This makes the reported compression ratio optimistic.

### A.8 COMPARISON OF FIXED BUDGET AND DYNAMIC BUDGET FOR R-KV AND H2O    NEW

In our evaluations, we adopt a dynamic budget strategy where each sample's budget is determined by its full length multiplied by the target sparsity, to ensure consistent compression ratios across samples. To illustrate the impact of this choice, we compare the performance of R-KV and H2O under both fixed and dynamic budget settings on Llama-3.1-8B-R1, Qwen-2.5-7B-R1, and Qwen-3-4B-Thinking across Math500 and AIME24 at sparsity levels of 0.2, 0.4, 0.6, and 0.8. In this comparison, the fixed budget per-sample is estimated as $budget(sample) = sparsity \times full\_length(sample)$, where $full\_length(sample)$ is the length of the response generated by the full KV cache model for that specific sample.

As shown in Figure 16, fixed-budget R-KV performs significantly worse than our dynamic-budget variant at 0.2, 0.4, and 0.6 sparsity, and only becomes better at 0.8 sparsity, while H2O maintains similar performance. This shows that our modification does not weaken the baselines; instead, it corrects an overly optimistic compression estimate and yields a more faithful comparison.

### A.9 DETAILED LATENCY MEASUREMENTS AND END-TO-END SPEEDUP    NEW

This section reports detailed per-layer latency measurements of attention with compressed KV cache by the head-reallocation diagram, and the end-to-end speedups of our simple PyTorch implementation.

Compared to the full model, the head-reallocation method needs to rearrange the Q, K, and V tensors into two dense groups in each attention computation: one for full-KV heads and one for compressed-KV heads. It then computes attention separately for these two groups and finally concatenates the

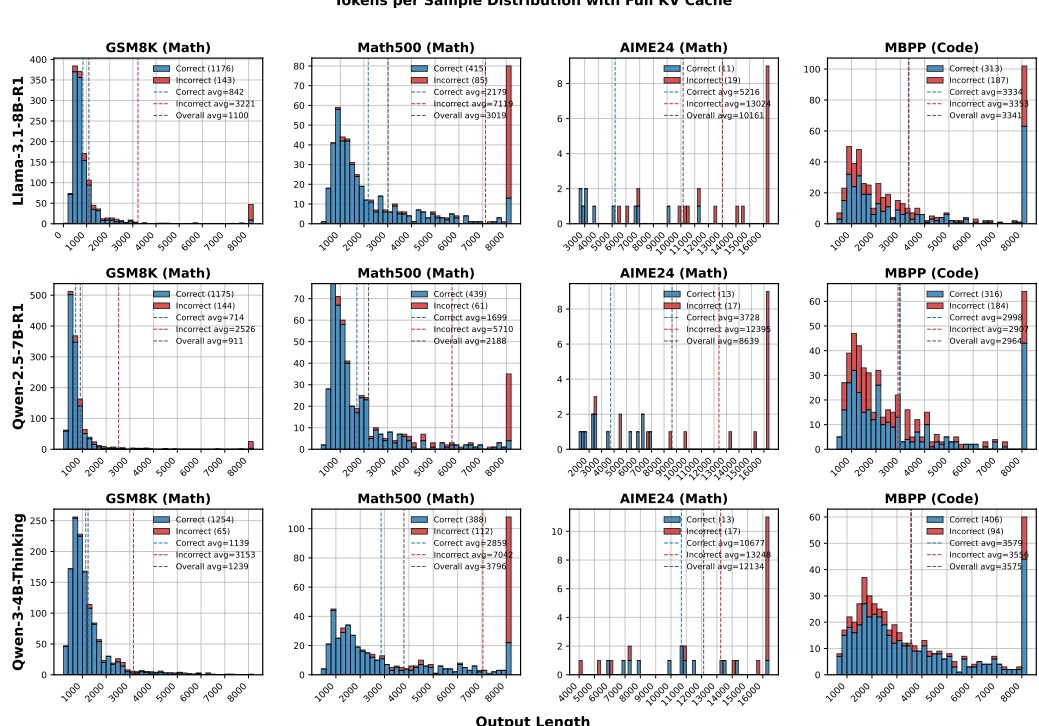

Figure 15: The distribution of output lengths on Math500 and AIME24 benchmarks with Llama-3.1-8B-R1, Qwen-2.5-7B-R1, and Qwen-3-4B-Thinking models with full KV cache.

outputs along the head dimension. This additional rearrangement and split computation introduces overhead.

We measure the latency of an attention forward with compressed attention by head reallocation on a single A800 GPU using the Llama-3.1-8B-R1 configuration with 32 attention heads, 8 KV heads, and a head dimension of 128. We randomly generate query, key, and value tensors to simulate attention computation with sequence lengths from 1K to 32K and sparsity levels from 0.1 to 0.8. The batch size is set to 128 for compressed attention, and for full attention, it is set to $\text{bound}(128 \times (1 - \text{sparsity}))$ to ensure similar memory consumption between the two methods. Throughput is calculated as $\text{throughput} = \text{batch\_size}/\text{latency}$. We use a PyTorch implementation with FlashAttention-2 for both full attention and compressed attention by head reallocation. Each configuration is run for 10 iterations with 3 warmup iterations, and the average latency is recorded.

We report the latency ratio (compressed / full) and throughput ratio (compressed / full) at a sequence length of 16K across different sparsity levels, and at a sparsity of 0.5 across all sequence lengths, as shown in Figure 17. For a fixed sequence length, as the compression ratio increases, the cost of head-wise operations approaches that of full attention and, under high compression and long sequences, can even be slightly lower. For a fixed sparsity, as the sequence length increases, the latency approaches that of full attention forward. Under a given memory budget, compression allows us to increase the batch size, and for sparsity above 0.2 this leads to throughput improvements. Given that our lossless compression typically lies in the 0.2-0.5 sparsity range, our current PyTorch implementation does not introduce prohibitive latency. We expect that a dedicated CUDA kernel for reorganizing the QKV tensors could further improve speed.

As for end-to-end speedup, we evaluate the impact of our method on serving latency using a standard PyTorch/Transformers inference pipeline with FlashAttention-2, without additional inference optimizations such as quantization or continuous batching. As shown in Table 7, we still observe end-to-end speedups when using RLKV.

The observed end-to-end speedups are smaller than the per-layer throughput gains reported above. A key reason is the large variation in output lengths for reasoning-style workloads, as shown in Fig-

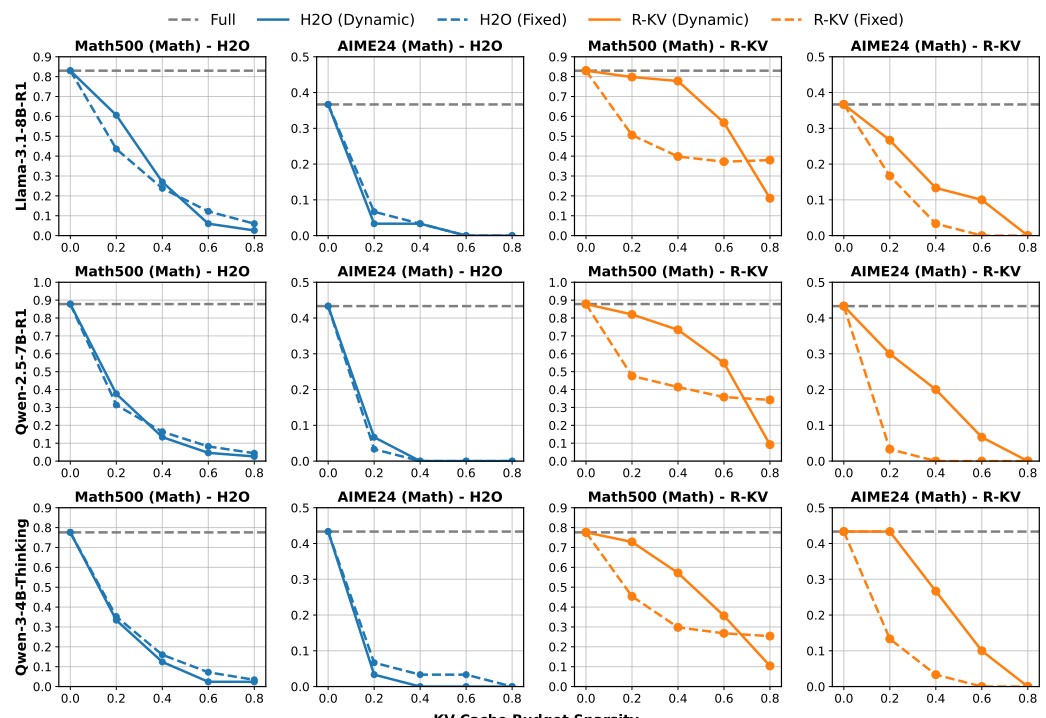

Figure 16: Performance comparison of R-KV and H2O under fixed budget and dynamic budget settings on Llama-3.1-8B-R1, Qwen-2.5-7B-R1, and Qwen-3-4B-Thinking across Math500 and AIME24 at sparsity levels of 0.2, 0.4, 0.6, and 0.8. The fixed-budget R-KV performs significantly worse than the dynamic-budget variant at 0.2, 0.4, and 0.6 sparsity, and only becomes better at 0.8 sparsity, while H2O maintains similar performance across both settings.

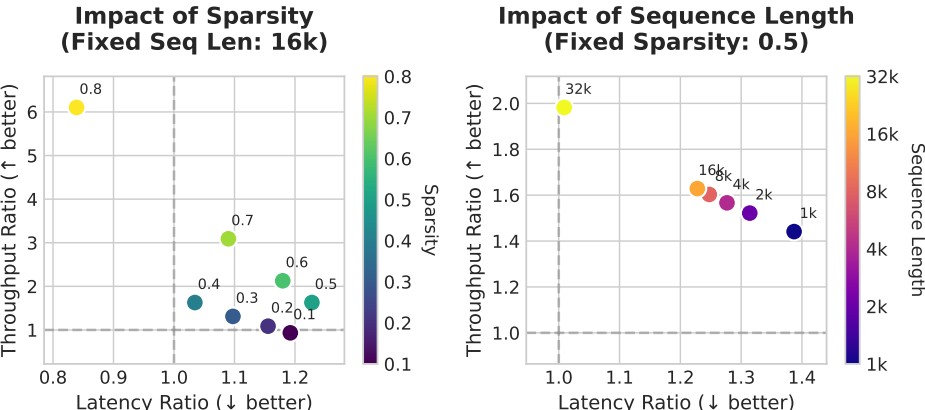

Figure 17: The latency of once attention computing of compressed attention by head-reallocation compared to full attention. *Left*: Varying sparsity levels at fixed sequence length of 16K. *Right*: Varying sequence lengths at fixed sparsity of 0.5.

ure 15: when requests in a batch terminate at different decoding steps, completed sequences remain in the batch and effectively waste compute. Modern inference frameworks such as SGLang (Zheng et al., 2024) and vLLM (Kwon et al., 2023) support continuous batching, where completed requests are removed and new requests are added to the batch on the fly, thereby reducing wasted computation due to heterogeneous output lengths. We expect that integrating head-reallocation attention into such frameworks could further improve end-to-end speedups.

Table 7: End-to-end serving metrics at sparsity 0.5 using a PyTorch/Transformers implementation. The table reports batch size, peak GPU memory, latency, speedup (normalized so the full model is 1.0), and accuracy for the full model and RLKV.

| # | Batch Size | | Peak GPU (GB) | | Latency (s) | | Speedup | | Accuracy | |
|---|---|---|---|---|---|---|---|---|---|---|
| | Full | RLKV | Full | RLKV | Full | RLKV | Full | RLKV | Full | RLKV |
| 1 | 2 | 4 | 19.08 | 19.40 | 24374.8 | 21080.2 | 1.00 | 1.16 | 0.810 | 0.792 |
| 2 | 4 | 8 | 23.57 | 23.84 | 16838.1 | 14569.2 | 1.00 | 1.16 | 0.784 | 0.792 |
| 3 | 8 | 16 | 32.23 | 32.82 | 14222.4 | 11767.5 | 1.00 | 1.21 | 0.776 | 0.768 |
| 4 | 16 | 32 | 49.79 | 50.88 | 11752.4 | 10809.1 | 1.00 | 1.09 | 0.770 | 0.764 |

