# OpenReview forum: "Which Heads Matter for Reasoning? RL-Guided KV Cache Compression"
_ICLR.cc/2026/Conference — Submitted to ICLR 2026_

### Official Review · Reviewer_HVGo · 2025-10-25

**Soundness:** 2
**Presentation:** 2
**Contribution:** 2
**Rating:** 2
**Confidence:** 4

**Summary:**

This paper addresses a head-level KV cache compression method specialized for reasoning tasks. The authors observe that previous approaches, such as the reasoning-specialized token-level KV dropping method (R-KV) and the factuality-focused head-level compression method (DuoAttention), exhibit significant performance degradation on reasoning tasks. To address this issue, the authors adopt DuoAttention’s optimization framework and apply their method to reasoning task datasets with reward signals. Since a naïve application of the method leads to unstable optimization, the authors propose several training stabilization techniques, which play a critical role in achieving strong compression performance. The effectiveness of the proposed head-level KV compression method is demonstrated on three math and one coding-based reasoning benchmark using the LLaMA3.1-8B-R1 and Qwen2.5-7B-R1 models. Benchmark results show that the proposed method consistently outperforms DuoAttention and R-KV, achieving a 20–50% reduction in KV cache memory for the evaluated benchmarks and models.

**Strengths:**

- The paper addresses a timely and relevant problem that is likely to attract the interest of the ICLR community.
- To resolve the training instability and degeneration issues, the authors introduce several training stabilization techniques based on their observations.
- The experimental results compare the proposed method with recent approaches and show consistent, albeit modest performance improvements.

**Weaknesses:**

- Misleading Figure 1 and main motivation
  - The figure refers to “Token-dropping” and “Head-reallocation”, but neither the figure nor the paper provides a fundamental explanation of these approaches. Instead, the paper focuses only on a specific implementation, including R-KV and DuoAttention. I recommend revising the figure and text to more accurately reflect the statement, as the current analysis refers not to general methodologies but to a specific implementation.

- Lack of conceptual justification for “reasoning heads”
  - The paper extends the concept of “retrieval heads” to propose the existence of “reasoning heads”, but this conceptual leap is not sufficiently justified. It is unclear (1) what exactly defines a “reasoning head”, and (2) the paper lacks fine-grained analysis to support its existence.
  - (1) Can we truly claim that specific heads perform a distinct reasoning function? For instance, in Figure 6, the so-called “reasoning heads” may be playing more general roles, such as supporting standard decoding or controlling the continuation of responses, rather than performing reasoning-specific functions. The paper does not sufficiently rule out these alternative explanations.
  - (2) Compared with previous studies such as “Retrieval Head Mechanistically Explains Long-Context Factuality” (2024), this paper does not provide multi-level (example-level, token-level) analyses for attention mechanism interpretation or comprehensive evaluations across diverse models and datasets. Without such detailed examination, the claim of “reasoning heads” remains unconvincing.

- Limited methodological or analytical novelty
  - Conceptually, the paper feels like a straightforward extension of DuoAttention to reasoning models, as it largely adopts DuoAttention’s architecture and optimization framework. The contribution does not substantially expand my understanding of attention mechanisms or yield surprising results.

**Questions:**

- In Figure 5, the results on LLaMA3.1–8B–R1 (GSM8K, math) and Qwen2.5–7B–R1 (MBPP, code) show very similar performance trends between DuoAttention and the proposed method. Why do these two methods behave similarly? This could indicate that the observed effects are not tied to reasoning-specific heads, but rather to more general/complex mechanisms or dataset-specific factors.

- Line 83: What is the meaning of “useless steps”? How are “useless” steps defined and identified in the experiments?

- In real-world applications, reasoning and factual retrieval often occur in a mixed manner. How would the proposed method handle such mixed scenarios? What level of lossless compression could be achieved in those cases?

---

> ### Author Response · Authors · 2025-11-23
> **Response to Reviewer HVGo (Part 1)**
>
> We sincerely thank the reviewer for their thoughtful feedback and constructive suggestions. Below are our responses to your concerns.
>
> ---
>
> **W1: Clarity of Figure 1**
>
> We recognize that the original Figure 1 was not sufficiently clear. In the revision, we have extended Figure 1 by adding a new panel (a), which provides a schematic illustration of the two general KV-cache compression strategies (token dropping and head reallocation), and we keep the empirical results as panel (b) as a case study. We have also revised the corresponding caption, as well as Paragraphs 2 and 3 in the introduction, for better clarity.
>
> R-KV and DuoAttention are currently SOTA methods within the token-dropping and head-reallocation methods respectively. Therefore, we believe that our analysis using them as representative case studies of these two strategies remains valuable.
>
> We hope this clarification makes our claims more precise.
>
> ---
>
> **W2: The Definition of Reasoning Heads**
>
> We thank the reviewer for reminding us not to overclaim. We emphasize that our claim does not involve a conceptual leap.
>
> * We define reasoning heads at line 96 in the first submission as:
>
>   > A subset of heads is critical to reasoning behaviors and naturally requires a full KV cache to maintain CoT consistency
>
>   This is an *operational* definition, and indicates the relationship between reasoning heads and KV cache access.
>
> * Reasoning behaviors are defined at line 33 in the first submission as:
>
>   > ... reasoning behaviors, such as self-reflection to revisit previous steps and exploration of alternative approaches, and achieve revolutionary performance on challenging mathematical and coding problems.
>
> * Since reasoning behaviors, by definition, revisit previous tokens, there must exist some heads whose attention scores require access to past KV cache states.
>
> * Our two case studies on R-KV and DuoAttention support this view.
>   R-KV suggests that reasoning models intrinsically require a full KV cache, so only head-reallocation can achieve compression without catastrophic failure. DuoAttention also exhibits performance drops, which motivates us to track the reasoning behaviors that the model produces during generation (Introduction, Paragraph 3).
>
> * RLKV attaches adapters to each head to quantify its dependence on a full KV cache. Under sparsity pressure in an RL framework, we optimize these adapters to learn a stable distribution. The reasoning behaviors appear in the generation process during RL sampling, while the reward signal directly evaluates their quality (Introduction, Paragraph 4).
>
> We do not claim that these reasoning heads perform a *unique* or *comprehensive* reasoning function. We apologize for any confusion our wording may have caused.
>
> * Given reasoning behaviors such as self-reflection to revisit previous steps and exploration of alternative approaches, it is natural that the heads that carry these behaviors need a full KV cache.
> * This does **not** mean that the remaining heads do not contribute to generation. Rather, they contribute under a *compressed* KV cache. This implies that they do not need to access the full historical state, and hence they do not contribute directly to reasoning behaviors. This is precisely the rationale that enables us to compress.
> * We also do not rule out additional functional roles behind reasoning heads. In fact, our second point in the future work section explicitly mentions:
>   > ... exploring the complete functional roles of these heads beyond reasoning could unlock new insights into model interpretability and architectural design.
> * We state that the reasoning heads learned by RLKV and the retrieval heads learned by DuoAttention are different. We never claim whether “reasoning heads” and “retrieval heads” are fundamentally the same or different concepts, because the retrieval heads in DuoAttention are obtained from its training objective—a synthetic recall task—which is itself an approximation.
>
> Compared with the paper [1] mentioned by the reviewer, our primary focus is *efficiency*: we study reasoning behaviors in reasoning models from the perspective of KV-cache compression. We view such a deeper functional study of the KV heads in reasoning models as future work, which is mentioned in our second point in the future work section.
>
> * During the rebuttal period, we added experiments on a third model, the very recent **Qwen-3-4B-Thinking**, shown in Appendix Section A.5.
> * We also added experiments on four subsets of the challenging knowledge QA benchmark suite MMLU-Pro: Chemistry, Computer Science, Law, and Physics, shown in Appendix Section A.6.
> * RLKV achieves competitive performance across **3 models and 8 evaluations** under various sparsity levels. We have tried our best to demonstrate the effectiveness of our method in a comprehensive way.
>
> [1] *Retrieval Head Mechanistically Explains Long-Context Factuality* (2024)

---

> ### Author Response · Authors · 2025-11-23
> **Response to Reviewer HVGo (Part 2)**
>
> **W3: Novelty**
>
> We appreciate the reviewer’s concern about novelty, and we agree that our original contribution paragraph may have overstated the conceptual claims and caused confusion. We have revised this paragraph in the introduction, and we re-clarify our main innovations compared to DuoAttention[1]:
>
> * First, the problem we study is different from DuoAttention. DuoAttention compresses long-context **in-context recall** tasks on the *instruction-tuned* models, while we focus on compressing **reasoning models** during long CoT generation.
>
> * Second, DuoAttention[1] and RozerAttention [2] both propose head-reallocation methods. Reasoning behaviors necessarily require a full KV cache, which naturally leads RLKV to adopt this paradigm.
>   RLKV introduces a different training objective tailored to reasoning. Instead of optimizing a proxy objective such as a synthetic recall task, RLKV learns per-head importance with an RL-based objective (RLVR) whose reward is computed by verifying the current policy’s own generated reasoning trajectories under sparsity. In this way, the allocation of full KV cache to specific heads is directly tied to reasoning performance, rather than to a separate retrieval-oriented proxy.
>
>
> * Third, our main results in Figure 5, the lossless performance in the tables, and the head-replacement experiments in Figure 6 all indicate that the heads identified by RLKV and DuoAttention differ. On some datasets and sparsity settings, RLKV outperforms the baseline by **10–20%** in accuracy.
>
> * Additionally, we compute the Spearman correlation between the importance scores of heads identified by RLKV and DuoAttention. We do not only observe weak correlations; more importantly, for the heads that each method considers highly dependent on a full KV cache, the distributions differ:
>
>   | Ratio              | top 100% | top 80% | top 60% | top 40% | top 20% |
>   | ------------------ | -------: | ------: | ------: | ------: | ------: |
>   | Llama-3.1-8B-R1    |    0.475 |   0.407 |   0.299 |   0.286 |   0.231 |
>   | Qwen-2.5-7B-R1     |    0.155 |   0.095 |   0.099 |  -0.182 |   0.045 |
>   | Qwen-3-4B-Thinking |    0.338 |   0.401 |   0.387 |   0.335 |   0.131 |
>
> [1] DuoAttention: Efficient Long-Context LLM Inference with Retrieval and Streaming Heads (2025)
> [2] RazorAttention: Efficient KV Cache Compression Through Retrieval Heads (2025)
>
> ---
>
> **Q1: Similar Accuracy curve**
>
> Appendix Section A.4, *Details of Error Modes Analyses*, already preliminarily presents the error modes of DuoAttention and RLKV in Figure 12.
>
> Although their performance degradation trends appear similar, their **error mode ratios** are different. We further examined our records:
>
> **Llama-3.1-8B-R1 on GSM8K**
>
> | sp  | # Opposite predictions | Duo-only correct | RLKV-only correct | # Mismatched error types |
> | --- | ---------------------- | ---------------- | ----------------- | ------------------------ |
> | 0.2 | 116                    | 55               | 61                | 8                        |
> | 0.4 | 168                    | 86               | 82                | 10                       |
> | 0.6 | 264                    | 121              | 143               | 29                       |
> | 0.8 | 517                    | 134              | 383               | 268                      |
>
> **Qwen-2.5-7B-R1 on MBPP**
>
> | sp  | # Opposite predictions | Duo-only correct | RLKV-only correct | # Mismatched error types |
> | --- | ---------------------- | ---------------- | ----------------- | ------------------------ |
> | 0.2 | 242                    | 114              | 128               | 17                       |
> | 0.4 | 235                    | 113              | 122               | 23                       |
> | 0.6 | 232                    | 103              | 129               | 81                       |
> | 0.8 | 8                      | 5                | 3                 | 176                      |
>
> The number of opposite predictions and the number of mismatched error types clearly demonstrate that the two methods behave differently, so their similarity in aggregate accuracy is not a mere coincidence.
>
> ---
>
> **Q2: The Meaning of "useless steps"**
>
> In the revised version, we have replaced the term *"useless steps"* with a direct description of the observed phenomenon:
>
> > For problems that the uncompressed model can solve within the maximum budget, the compressed model goes astray in its reasoning process and is unable to reach a correct solution before the budget is exhausted.
>
> We hope this wording is clearer and avoids potential misunderstanding.

---

> ### Author Response · Authors · 2025-11-23
> **Response to Reviewer HVGo (Part 3)**
>
> **Q3: The Mixed Manners of Reasoning and Retrieval**
>
> We fully agree that real-world usage often mixes reasoning and factual retrieval. In fact, our evaluation records already exhibit this mixture: solving math or coding problems frequently requires recalling earlier definitions, intermediate results, or parts of the problem statement, and revisiting them during self-reflection.
> It suggests that reasoning behaviors in practice often involve varying degrees of retrieval rather than being isolated from it.
>
> Methodologically, RLKV does not exclude retrieval behaviors during learning, which heads should be allocated the full KV cache. Retrieval behaviors also require a full KV cache naturally.
> We suggest that RLKV guiding KV cache compression would still be effective in the real-world mixed scenarios. Unlike synthetic recall tasks (which have clearly defined retrieval metrics) or standard reasoning benchmarks (which have verifiable final answers), real-world applications can be harder to evaluate in a controlled way. We view the construction of such benchmarks as an important direction for future work. We believe this would be very helpful for deepening our understanding of the internal mechanisms of reasoning models.
>
> Notably, in the revision, we additionally evaluate RLKV and baselines on four knowledge-domain MMLU-Pro subsets (Chemistry, Computer Science, Law, Physics), where retrieval behaviors may play a larger role, and RLKV still achieves competitive performance (Appendix Section A.6). We believe this may partially alleviate the reviewer’s concerns.
>
> ---
>
> We hope our response has clarified your concerns and can improve your rating of our paper.

---

### Official Review · Reviewer_Sm12 · 2025-10-28

**Soundness:** 3
**Presentation:** 3
**Contribution:** 3
**Rating:** 4
**Confidence:** 4

**Summary:**

The paper studies KV-cache compression for reasoning LLMs and argues that heads differ functionally in this regime: a small subset of “reasoning heads” must retain full KV cache to preserve chain-of-thought (CoT) integrity, while other heads can use a compressed cache. It proposes RLKV, which inserts per-head gating adapters that mix full attention with streaming/local attention; the adapters are trained by group-relative PPO/GRPO on verifiable reasoning tasks, with an L1 sparsity penalty to push most heads toward compressed access.

**Strengths:**

1.The problem formulation is right and fresh. It clearly shows why long CoTs break existing compression: token dropping induces repetition/loops; retrieval-head reallocation preserves some integrity but derails reasoning steps.

2. The performance of the proposed method is impressive. It shows consistent gains over H2O, R-KV, DuoAttention across tasks and sparsities; even beats full KV on AIME24 at certain budgets, suggesting noise from non-reasoning heads.

3. The memory reduction is clear. The paper reports 20–50% KV savings at near-lossless quality

**Weaknesses:**

1. The head identity stability is not discussed. The importance of head may depend on prompt style/length and reasoning mode; the paper shows global head rankings, but does not analyze input-conditional head selection or per-layer diversity across tasks.

2. The end-to-end acceleration is not reported. We get memory reductions, but not end-to-end speedups under realistic batching/servers with paged KV, quantized KV, etc. The head-wise design may not improve the end-to-end latency, since the latency can be blocked by other heads in the same layer.

**Questions:**

Corresponding to the weaknesses, the following questions are not answered in the paper:

1. If you train gates on math, how much zero-shot transfer do you get to coding or to multi-hop QA? Conversely, do gates trained on MBPP hurt math?

2. To accelerate the inference speed, is it possible to compress the KV in a layer-wise? or the current head-wise design already accelerated the inference speed greatly?

---

> ### Author Response · Authors · 2025-11-23
> **Response to Reviewer Sm12 (Part 1)**
>
> We sincerely thank the reviewer for their thoughtful feedback and constructive suggestions. Below are our responses to your concerns.
>
> ---
>
> **W1: Head Identity Stability**
>
> We understand the reviewer’s concern about whether the distribution of reasoning heads may change under different input conditions.
>
> First, we clarify our definition of reasoning heads, which require the full KV cache access to preserve reasoning behaviors. The remaining heads are not removed or disabled: they still contribute to the attention computation with a compressed constant-size KV cache instead of the full KV cache.
>
> Functionally, these remaining heads continue to operate using the limited history stored in the compressed KV cache during inference. If our identification of reasoning heads is accurate, then under various input conditions, the model should preserve final-answer accuracy comparable to the full model. This is exactly what we observe on benchmarks outside the training domain: RLKV achieves competitive performance on code benchmarks and on the four subsets of the MMLU-Pro knowledge benchmark we added during the rebuttal period (Appendix Section A.6). Moving from math to code and knowledge-intensive QA already constitutes a substantial shift in input conditions, and the fact that a single **static** head distribution continues to perform strongly under this shift provides strong empirical evidence of robustness.
>
> Moreover, the RLKV training process is quite strict. We train on 3k math problems with varying difficulty, and a trajectory receives a reward **only when the final answer is correct**. This strongly enforces that the identified head distribution truly reflects each head’s need for full KV access. From the model’s perspective, for a given math problem, the correct final answer should remain invariant even when input conditions vary, so the learned distribution of reasoning heads must be robust to such variations.
>
> Regarding prompt style, we follow the prompt setting (output-format instruction + question) recommended by DeepSeek-R1 [1] in both training and evaluation without additional prompt engineering. DeepSeek-R1 reports that few-shot prompting consistently degrades its performance. This zero-shot prompt setting is consistent across all our experiments, so the identified reasoning heads are not tied to a specific prompt style. The detailed prompt format will be updated in Appendix Section A.1.
>
> [1] DeepSeek-R1: Incentivizing Reasoning Capability in LLMs via Reinforcement Learning

---

> ### Author Response · Authors · 2025-11-23
> **Response to Reviewer Sm12 (Part 2)**
>
> **W2: Head-Wise Operations and Attention Latency**
>
> We understand the reviewer’s concern that memory reduction alone is not sufficient, and that latency is also critical. We acknowledge that the head-wise operations required to support the compressed KV cache introduce additional computation and thus extra latency. We update the details in Appendix Section A.9. Our response addresses this from three aspects:
>
> **(1) Per-layer attention latency**
>
> * **For fixed sequence length**, as the compression ratio increases, the cost of head-wise operations approaches that of full attention, and under high compression and long sequences, it can even be faster than full attention (e.g., at 8k with 0.8 sparsity, ratio 0.904; at 32k with 0.1 sparsity, ratio 0.986).
> * **For fixed sparsity**, as sequence length increases, the latency approaches that of full attention.
> * Under a given memory budget, compression allows us to increase batch size. For sparsity ≥ 0.2, this leads to **throughput improvements**. Given that our lossless compression regime typically lies in the 0.2–0.5 sparsity range, our current PyTorch implementation already does not introduce prohibitive latency. We expect that a dedicated CUDA kernel implementation could further improve speed.
>
> Below, we report some of the measured latency and throughput ratios (compressed vs. full):
>
> **Sparsity = 0.2**
>
> | Seq Len          | 1K        | 2K        | 4K        | 8K        | 16K       | 32K       |
> | ---------------- | --------- | --------- | --------- | --------- | --------- | --------- |
> | Batch Size       | 128 / 102 | 128 / 102 | 128 / 102 | 128 / 102 | 128 / 102 | 128 / 102 |
> | Latency Ratio    | 1.231     | 1.199     | 1.187     | 1.168     | 1.155     | 0.945     |
> | Throughput Ratio | 1.019     | 1.047     | 1.058     | 1.074     | 1.086     | 1.329     |
>
> **Sparsity = 0.5**
>
> | Seq Len          | 1K       | 2K       | 4K       | 8K       | 16K      | 32K      |
> | ---------------- | -------- | -------- | -------- | -------- | -------- | -------- |
> | Batch Size       | 128 / 64 | 128 / 64 | 128 / 64 | 128 / 64 | 128 / 64 | 128 / 64 |
> | Latency Ratio    | 1.387    | 1.314    | 1.277    | 1.248    | 1.228    | 1.009    |
> | Throughput Ratio | 1.441    | 1.522    | 1.567    | 1.603    | 1.629    | 1.982    |
>
> **Seq Len = 4K**
>
> | Sparsity         | 0.1       | 0.2       | 0.3      | 0.4      | 0.5      | 0.6      | 0.7      | 0.8      |
> | ---------------- | --------- | --------- | -------- | -------- | -------- | -------- | -------- | -------- |
> | Batch Size       | 128 / 115 | 128 / 102 | 128 / 89 | 128 / 76 | 128 / 64 | 128 / 51 | 128 / 38 | 128 / 25 |
> | Latency Ratio    | 1.209     | 1.187     | 1.130    | 1.082    | 1.277    | 1.233    | 1.182    | 1.025    |
> | Throughput Ratio | 0.921     | 1.058     | 1.272    | 1.557    | 1.567    | 2.035    | 2.850    | 4.992    |
>
> **Seq Len = 16K**
>
> | Sparsity         | 0.1       | 0.2       | 0.3      | 0.4      | 0.5      | 0.6      | 0.7      | 0.8      |
> | ---------------- | --------- | --------- | -------- | -------- | -------- | -------- | -------- | -------- |
> | Batch Size       | 128 / 115 | 128 / 102 | 128 / 89 | 128 / 76 | 128 / 64 | 128 / 51 | 128 / 38 | 128 / 25 |
> | Latency Ratio    | 1.192     | 1.155     | 1.097    | 1.034    | 1.228    | 1.180    | 1.090    | 0.839    |
> | Throughput Ratio | 0.934     | 1.086     | 1.311    | 1.628    | 1.629    | 2.127    | 3.091    | 6.103    |
>
> **(2) End-to-end speedup**
>
> In end-to-end serving, the overall pipeline is much more complex than a single attention layer. Nonetheless, under a straightforward PyTorch/Transformers implementation, we **still observe speedups**. Our method is orthogonal to the techniques mentioned by the reviewer, and combining them should provide further acceleration.
>
> Below, we report end-to-end results for 0.5 sparsity:
>
> |Batch Size (compressed / full)|4/2|8/4|16/8|32/16|
> |-|-|-|-|-|
> |Peak GPU Mem (GB)|19.40/19.08|23.84/23.57|32.82/32.23|50.88/49.79|
> |Latency (S)|21080.2/24374.8|14569.2/16838.1|11767.5/14222.4|10809.1/11752.4|
> |Speedup Ratio|1.16|1.16|1.21|1.09|
> |Acc.|0.792/0.810|0.792/0.784|0.768/0.776|0.764/0.770|
>
> We note that the end-to-end speedups are smaller than the per-layer **throughput ratios**. One key reason is the **diverse output lengths** of reasoning models, as shown in Figure 15 in Appendix Section A.7. When batched requests terminate at different steps, completed samples effectively waste compute in this native implementation.
> Modern inference frameworks such as sglang and vLLM support *continuous batching*, where completed requests are removed and new requests are added to the batch on the fly. We expect that integrating head-reallocation attention into such frameworks could further improve end-to-end speedups. Due to time and engineering constraints, we were not able to report such results in the rebuttal. We plan to add the support to sglang when we open-source our code.

---

> ### Author Response · Authors · 2025-11-23
> **Response to Reviewer Sm12 (Part 3)**
>
> **Q1: Training Domain and Generalization**
>
> We understand the reviewer’s concern about training-domain specificity. As we already discussed in **W1**, our prompt setting consists only of a format instruction followed by the question, which is inherently a **zero-shot** setup. Our reported MBPP scores are also obtained in the zero-shot setting.
>
> To further evaluate generalization beyond the math training domain, we additionally report four subsets of the challenging knowledge QA benchmark suite MMLU-Pro: Chemistry, Computer Science, Law, and Physics. The detailed results are given in Appendix Section A.6.
>
> We choose the math domain for RL training primarily because high-quality open-source math datasets are abundant and cover a wide range of difficulties. This allows us to sample 3k problems, each inducing trajectories from roughly 1k to 7k tokens from the full model, which is crucial for our RL setup.
>
> Concretely, we did attempt to follow the reviewer’s suggestion to train on code, but existing open-source code datasets are not ideal for our RL setting. For example, the publicly available MBPP set only contains around 1k problems without diverse difficulty levels, which led to weak reward signals in our preliminary experiments. We also examined PRIME-RL/Eurus-2-RL-Data, but on this dataset, the base model’s accuracy is essentially 0% on 21k examples, so the reward is almost always zero and the RL signal quickly collapses.
>
> We believe that **appropriate code datasets or other QA datasets**, if available, could be used similarly: they would allow the model to generate trajectories of varying lengths, and RL could then use the final answer to compute rewards. In this way, the RLKV procedure could also identify reasoning heads that require access to the full KV cache in non-math domains. At present, however, high-quality open-source math data is the only domain where such a setup is readily feasible.
>
> ---
>
> **Q2: Feasibility of a Layer-wise Design**
>
> We agree that a layer-wise design would reduce the head-wise operation overhead discussed in **W2** by avoiding per-head KV reallocation. Under such a design, some layers would use a purely streaming attention pattern, while others would use full attention.
>
> However, as shown in Figure 3 of our paper, the distribution of reasoning heads identified by RLKV **does not exhibit strong layer-wise consistency**. Reasoning heads are scattered across layers rather than concentrated in a small subset of them. A strict layer-wise design would therefore force us to treat entire layers as full or streaming, which we expect to severely harm performance.
>
> Our current head-wise reallocation design has an additional advantage: within the same layer, the value vectors from full-attention heads and streaming-attention heads can interact through the shared projection $W_\text{proj}$, which may be crucial for preserving reasoning behavior.
>
> This directly addresses the question about whether a layer-wise KV compression scheme could replace our head-wise design; we evaluate a **layer-wise RLKV** variant on Math500 with Llama-3.1-8B. We obtain a layer-wise distribution by averaging the reasoning-head distribution within each layer and then assigning full KV to the entire layer accordingly. The results are:
>
> | Sparsity        | Full  | 0.2   | 0.4   | 0.6   | 0.8   |
> | --------------- | ----- | ----- | ----- | ----- | ----- |
> | Layer-wise RLKV | 0.830 | 0.776 | 0.724 | 0.470 | 0.120 |
> | Head-wise RLKV  | 0.830 | 0.844 | 0.846 | 0.780 | 0.496 |
>
> The layer-wise variant suffers **substantial performance degradation**, especially at higher sparsities. In the face of such large accuracy drops, the additional speedup from eliminating head-wise operations becomes much less meaningful. This empirical result supports our design choice of using head-wise rather than layer-wise reallocation.
>
> ---
>
> We hope that these clarifications address the reviewer’s concerns and help convey both the robustness and practicality of RLKV. We would be grateful if these additional analyses and experimental results could lead to a more favorable assessment of our work.

---

### Official Review · Reviewer_Y94n · 2025-10-30

**Soundness:** 2
**Presentation:** 3
**Contribution:** 2
**Rating:** 4
**Confidence:** 3

**Summary:**

This work addresses the critical challenge of KV cache compression for LLMs specialized for reasoning. The authors posit that existing compression techniques (both token-dropping and head-reallocation) fail on these models because the extended CoT generation process introduces a new functional requirement not previously addressed. The core hypothesis is that reasoning models exhibit "functional heterogeneity", and the paper introduces the concept of "reasoning heads". To identify these heads, the authors propose the RLKV framework, which uses RL to directly optimize downstream reasoning quality. The RL agent is trained using a verifiable reward signal (final answer correctness on math problems) and an $L1$ penalty to encourage sparsity. Empirically, the authors claim SOTA performance, achieving 20-50% KV cache reduction on models like Llama-3.1-8B-R1 and Qwen-2.5-7B-R1 with "near-lossless" performance, significantly outperforming baselines.

**Strengths:**

1.  The paper insightfully identifies that the primary bottleneck in modern reasoning models is not just long-context *ingestion* (which many KV cache methods target) but long-context *generation* from CoT. The motivational analysis in Figure 1, which contrasts instruct vs. reasoning models on the same task, is a strong validation of this problem statement.
2.  The main conceptual contribution is the shift in optimization objective. Instead of relying on proxy metrics like next-token loss or output deviation, RLKV directly optimizes the true downstream task metric—reasoning accuracy—via an RL reward. Optimizing for semantic correctness rather than syntactic similarity is a significant idea.
3.  The authors identify the inherent conflict between a sparse, unstable task reward and a dense, stable $L1$ regularization penalty. The proposed solutions in Section 3.3 (adaptive penalty weights and curriculum-based data sampling) are clever, well-reasoned, and appear effective for making the RL training viable, as demonstrated in the ablation studies.

**Weaknesses:**

1.  The paper's claims of generalization are not supported by the data and reflect a misunderstanding of the benchmark landscape. RLKV is trained on 3,000 mathematical reasoning problems from the DeepScaleR dataset. The authors then claim that MBPP (a code generation benchmark) serves as an evaluation of "out-of-distribution generalization". This may not be correct; mathematics (like AIME, DeepScaleR) and code (like MBPP, HumanEval) are potentially considered to be *related, in-domain reasoning tasks*. Therefore, the paper should provide more out-of-distribution generalization tests (e.g., to narrative summarization, logical planning, or RAG-based QA).

2.  For a paper proposing an *efficiency* method, the absence of a training cost analysis is a critical omission.
- The paper proposes a new, expensive RL training phase. RL for LLMs is known to be computationally intensive and complex to scale.
- This method is benchmarked against alternatives that are either training-free (H2O, R-KV) or have significantly cheaper, non-RL training (e.g., DuoAttention's single-pass proxy task or PruLong's next-token loss).
- The authors have shifted a large, undisclosed amount of computation from inference to this new training stage. Without a "Total Cost" analysis (Training FLOPs + Inference FLOPs), it is impossible to assess if RLKV provides practical efficiency gains.

3.  As admitted in Appendix A.2, the authors "convert fixed budgets to dynamic allocation" for H2O and R-KV. But these methods are explicitly designed as fixed-budget algorithms. This modification means the authors may not be comparing against the SOTA baselines as published, invalidating the empirical claims in Figure 5 and Table 1.

4.  The text states "...primarily due to quadratic attention computation and expanding KV cache." This is incorrect. The KV cache memory grows *linearly* ($O(n)$) with sequence length, not quadratically.

5.  Typo: "AIME25" in the appendix vs. "AIME24" in the main body.

**Questions:**

See weaknesses.

---

> ### Author Response · Authors · 2025-11-23
> **Response to Reviewer Y94n (Part 1)**
>
> We sincerely thank the reviewer for their thoughtful feedback and constructive suggestions. Below are our responses to your concerns.
>
> ---
>
> **W1: Clarify the OOD Evaluation**
>
> We understand the reviewer’s concern. In the revised version, we now use the phrase *"generalization beyond the training domain"* at line 315 instead of the original *"out-of-generalization generalization"*. Generalization on code benchmarks remains important for our goal: RLKV identifies those reasoning heads that can access the full KV cache, so we need to ensure they do not overfit to the math domain.
>
> We also agree with the suggestion to expand beyond math and code, and we have added results on four subsets of the challenging knowledge QA benchmark suite MMLU-Pro: Chemistry, Computer Science, Law, and Physics.
>
> The following table reports the average accuracy across the four subsets for different sparsity levels:
>
> | Llama-3.1-8B-R1 | full  | 0.2       | 0.4       | 0.6       | 0.8       |
> | --------------- | ----- | --------- | --------- | --------- | --------- |
> | RLKV            | 0.436 | **0.421** | **0.411** | **0.371** | 0.036     |
> | DuoAttn         | 0.436 | **0.421** | 0.384     | 0.278     | **0.050** |
> | R-KV            | 0.436 | 0.384     | 0.325     | 0.242     | 0.044     |
> | H2O             | 0.436 | 0.312     | 0.155     | 0.040     | 0.000     |
>
> | Qwen-2.5-7B-R1 | full  | 0.2       | 0.4       | 0.6       | 0.8       |
> | -------------- | ----- | --------- | --------- | --------- | --------- |
> | RLKV           | 0.458 | **0.461** | **0.434** | **0.329** | 0.015     |
> | DuoAttn        | 0.458 | 0.414     | 0.386     | 0.226     | **0.029** |
> | R-KV           | 0.458 | 0.385     | 0.315     | 0.191     | 0.010     |
> | H2O            | 0.458 | 0.239     | 0.135     | 0.049     | 0.011     |
>
> As shown, RLKV demonstrates strong generalization beyond the math domain: it matches or outperforms the baselines at sparsity levels 0.2–0.6.
> These results, detailed in Appendix Section A.6, further support that the heads identified by RLKV support general reasoning behaviors rather than only math-specific behaviors.
>
> The design of RLKV is not tied to mathematics: it learns which KV heads access the full KV cache to support reasoning behaviors. Thus, it is expected that these heads also contribute to reasoning in other domains, such as knowledge QA.
>
> While the additional MMLU-Pro subsets we include are not long-context or RAG-style benchmarks, they do move beyond math/code, which is closer in spirit to the reviewer’s suggested tasks than our original training domain. We agree that such long-context evaluations are valuable, and we see extending RLKV to these settings as important future work; however, due to time constraints, we were unable to include them in this revision.
>
> ---
>
> **W2: Training Cost vs. Inference Efficiency**
>
> We thank the reviewer for raising the efficiency concern and clarify the following.
>
> **Reported Training Cost.**
> We do report the training cost of RLKV and will make it more prominent in the revised version. Concretely:
>
> * Llama-3.1-8B-R1: 40 GPU-hours on 2×A100 (DuoAttention: 24 GPU-hours on 4×A100)
> * Qwen-2.5-7B-R1: 22 GPU-hours on 2×A100 (DuoAttention: 16 GPU-hours on 4×A100)
> * Qwen-3-4B-Thinking: 36 GPU-hours on 2×A100 (DuoAttention: 20 GPU-hours on 4×A100)
>
> We clarify that RLKV focuses on inference efficiency: throughout the paper, we consistently present RLKV as guiding efficient inference via KV cache compression. RLKV follows a standard “train-once, compress-forever” paradigm, similar to QAT [1] or structured pruning [2], where a one-time compression training cost is amortized over all future inference.
>
> **Lightweight RL and Practicality.**
> RLKV is not a full-model RLHF/RLVR: the base LLM is frozen, and we train only lightweight gating adapters on attention heads. This greatly reduces memory requirements; the main cost arises from rollouts. We further employ self-distillation to select 3k training problems, which keeps the overall training time modest.
>
> **Inference-Centric Efficiency and Total Cost.**
> RLKV is explicitly designed as an inference-efficiency method via KV cache compression. Once trained, it consistently achieves 20–50% KV cache reduction with near-lossless reasoning performance, and this benefit applies to all subsequent inference.
>
> We fully agree that understanding the overall training–inference tradeoff is important in practice. Currently, there is no widely accepted unified "total cost" metric in the community. In principle, this should be written as `total_cost = training_cost + n * inference_cost`,
> but the choice of `n` is inherently deployment-specific. As a result, any single scalar "total cost" depends on assumptions about the target application, which can vary widely across use cases.
>
> [1] Quantization and Training of Neural Networks for Efficient Integer-Arithmetic-Only Inference (2018)
> [2] Learning both Weights and Connections for Efficient Neural Networks (2015)

---

> ### Author Response · Authors · 2025-11-23
> **Response to Reviewer Y94n (Part 2)**
>
> **W3: An Implicit Unfair Comparison in the Fixed-Budget Setting**
>
> We agree that more clarity is needed on why we adopt a dynamic budget for R-KV and H2O.
>
> - Prior long-context KV compression works typically evaluate on in-context recall tasks, where each sample’s prompt length is fixed/controlled. A fixed budget of the form `budget = sparsity × prompt_length` then yields a roughly consistent compression ratio per sample, so fixed budgets are fair in that setting.
>
> - For reasoning tasks, however, the response length (CoT) is often much larger than the prompt. If we use a *global* fixed budget (e.g., 1k tokens), any sample whose full output fits within 1k tokens is **uncompressed**, while longer samples are compressed. Thus, different samples experience very different compression ratios, and fixed budgets are no longer fair at the per-sample level.
>
> - By measuring the full-model output lengths, we observe that they are highly diverse. For example, on Math500, the average length for correct samples is ≈2k tokens and for incorrect samples is ≈5k tokens, with an overall average of ≈3k tokens.
>
> - In R-KV, the reported compression rate is computed as `budget / average_full_length`. With a fixed budget of 2k and an average full length of 3k, this yields a nominal compression ratio of ≈69%.
>     However, a large fraction of samples are uncompressed, which were generated as the same as the correct responses of the full model.
>     This makes the reported compression ratio optimistic.
>
> - To keep the **fair** compression rate identical across methods and samples, we adopt a per-sample fixed budget:`budget(sample) = sparsity * full_length(sample)`, where `full_length(sample)` is measured from the uncompressed model and serves as an estimate of the final output length.
>
> Under this fair per-sample fixed budgeting scheme, R-KV performs significantly worse than our dynamic-budget variant at 0.2–0.6 sparsity, and only becomes better at 0.8 sparsity, while H2O maintains similar performance.
>
> |Llama-3.1-8B-R1 Math500|full|0.2|0.4|0.6|0.8|
> |-|-|-|-|-|-|
> |Fixed R-KV|0.830|0.506|0.398|0.372|0.380|
> |Dynamic R-KV|0.830|0.798|0.778|0.568|0.188|
> |Fixed H2O|0.830|0.436|0.238|0.122|0.060|
> |Dynamic H2O|0.830|0.606|0.270|0.060|0.026|
>
> |Llama-3.1-8B-R1 AIME24|full|0.2|0.4|0.6|0.8|
> |-|-|-|-|-|-|
> |Fixed R-KV|0.367|0.167|0.033|0.000|0.000|
> |Dynamic R-KV|0.367|0.267|0.133|0.100|0.000|
> |Fixed H2O|0.367|0.067|0.033|0.000|0.000|
> |Dynamic H2O|0.367|0.033|0.033|0.000|0.000|
>
> This shows that our modification does not weaken the baselines; instead, it corrects an overly optimistic compression estimate and yields a more faithful comparison.
>
> In response, we have updated Appendix Section A.7 and A.8 to provide more comprehensive details about the analysis of output lengths of full models and more results of the fixed-budget vs. dynamic-budget evaluations on Qwen-2.5-7B-R1 and Qwen-3-4B-Thinking.
>
> ---
>
> **W4: Quadratic Cost of Attention Computation**
>
> Our original statement in W4 refers only to the quadratic cost of *attention computation*, which is inherently quadratic in sequence length, and does not describe the KV cache. To avoid any ambiguity, we have updated the wording as follows:
>
> > “…due to the quadratic attention computation and the **linearly** expanding KV cache.”
>
> ---
>
> **W5: Typo in Appendix**
>
> Thank you for pointing this out. The correct benchmark name is AIME24, and we have fixed the typo in the appendix.
>
> ---
>
> We hope our response has clarified your concerns and can improve your rating of our paper.

---

### Official Review · Reviewer_vzQQ · 2025-10-30

**Soundness:** 4
**Presentation:** 3
**Contribution:** 4
**Rating:** 8
**Confidence:** 5

**Summary:**

The paper presents RLKV, an RL–based framework for reasoning-aware KV cache compression in LLMs.  The key insight is that attention heads in reasoning models are functionally heterogeneous, with only a subset being essential for maintaining CoT. RLKV introduces mixed attention with learnable gating adapters and optimizes them via GRPO using reasoning accuracy as the reward and L1 sparsity regularization to identify which heads truly matter. The method adaptively allocates full KV cache to reasoning-critical heads and compressed cache to others, achieving 20–50% memory reduction while maintaining near-lossless or even improved reasoning performance on multiple benchmarks such as GSM8K, Math500, AIME24, MBPP, and models such as Llama-3.1-8B-R1 and Qwen-2.5-7B-R1. Extensive analyses show that reasoning heads are more crucial than other heads, and the proposed adaptive penalty and self-distillation techniques stabilize RL training. Overall, RLKV provides a novel and interpretable approach to efficient reasoning inference.

**Strengths:**

1. The motivation that only a subset of attention heads is critical for reasoning is very interesting. It's also well supported by empirical evidence. The observation that near-lossless reasoning performance can be maintained even when 50–80% of KV heads are compressed makes this hypothesis convincing.
2. The method does not require model retraining and instead learns a small number of gating parameters to mix full and compressed attention. This design choice makes the approach easy to adopt in practice.
3. The method achieves strong results across multiple reasoning benchmarks, maintaining almost full accuracy with 20–50% KV cache reduction.
4. The paper provides clear evidence for the functional role of reasoning heads, with ablation and masking studies showing they are substantially more important than retrieval heads.

**Weaknesses:**

1. As mentioned in the paper, some heads are more important for reasoning than others, but their original functional roles in the base models (e.g., Qwen2.5-Math-7B-Base) remain unclear. It would strengthen the paper to analyze whether these reasoning-critical heads are specialized in math or coding-related attention patterns in the base model. A suggested experiment is to apply the same compression scheme to the base model and evaluate its performance on reasoning and general tasks (as in Figure 6), which could reveal whether reasoning heads emerge during post-training or pre-exist in base architectures.
2. In Figure 3, most gating coefficients (α) in Qwen-2.5-7B-R1 appear close to 1, suggesting the model heavily favors full attention across heads. This trend seems inconsistent with the sparsity objective.
3. The current method freezes model weights. Is it possible/feasible to explore joint training of gates and model parameters, which may yield stronger compression performance?

**Questions:**

Please see weaknesses.

---

> ### Author Response · Authors · 2025-11-23
> **Response to Reviewer vzQQ**
>
> We sincerely thank the reviewer for their thoughtful feedback and constructive suggestions. Below are our responses to your concerns.
>
> ---
>
> **W1: Head Importance Analyses on Base Models**
>
> We agree with the reviewer’s suggestion to extend the head-importance analyses to three base models (Llama-3.1-8B [1], Qwen-2.5-Math-7B [2], Qwen-3-4B-Base [3]) under the Math500 benchmark.
>
> We progressively replaced the top 10%, 20%, 30%, and 40% of attention heads with compressed KV cache, either (1) chosen at random or (2) selected according to the RLKV-learned gating values, and evaluated the resulting performance degradation.
>
> For **Llama-3.1-8B**, the base model performs poorly on reasoning tasks and often fails to produce well-formatted, verifiable outputs:
>
> |Llama-3.1-8B|full|0.1|0.2|0.3|0.4|
> |-|-|-|-|-|-|
> |random|0.068|0.064|0.070|0.068|0.060|
> |RLKV|0.068|0.054|0.046|0.048|0.040|
>
> For both **Qwen-2.5-Math-7B** and **Qwen-3-4B-Base**, the heads identified by RLKV have a substantially larger impact on reasoning accuracy than randomly replaced heads, even at the base-model stage:
>
> |Qwen-2.5-Math-7B|full|0.1|0.2|0.3|0.4|
> |-|-|-|-|-|-|
> |random|0.710|0.730|0.666|0.654|0.442|
> |RLKV|0.710|0.666|0.534|0.290|0.122|
>
> |Qwen-3-4B-Base|full|0.1|0.2|0.3|0.4|
> |-|-|-|-|-|-|
> |random|0.754|0.738|0.736|0.668|0.554|
> |RLKV|0.754|0.676|0.504|0.304|0.214|
>
> However, we still cannot claim that the functional roles of these heads already exist in the base models before post-training.
> Our operational definition of “reasoning heads” is that they require access to the full KV cache to support reasoning behaviors.
> The above experiment only shows that these heads in the base models also rely on the full KV cache; since the base models do not yet exhibit rich and complex reasoning behaviors, we cannot conclude that they implement the same reasoning functions as in the post-trained models.
> Clarifying how these functional roles emerge remains a valuable direction for future work.
>
> [1] The Llama 3 Herd of Models (2024)
> [2] Qwen2.5-Math Technical Report: Toward Mathematical Expert Model via Self-Improvement (2024)
> [3] Qwen3 Technical Report (2025)
>
> ---
>
> **W2: Less Sparse Distribution in Qwen-2.5-7B-R1**
>
> We acknowledge that the adapter values in Qwen-2.5-7B-R1 appear close to 1.
>
> However, this is **not** inconsistent with our sparsity objective. RLKV balances performance and sparsity, and for this model, pushing adapters further toward zero dramatically destroys its reasoning behavior. Thus, the optimizer sparsifies the heads only to the extent that the model can still maintain performance.
>
> This behavior is primarily driven by architectural differences. Qwen-2.5-7B-R1 contains substantially fewer KV heads and a larger KV group size, meaning each head carries more functional load.
>
> The architectural statistics below highlight this contrast:
>
> |Model|num_layers|num_kv_heads|total_num_kv_heads|kv_group_size|mean|std|
> |-|-|-|-|-|-|-|
> |Llama-3.1-8B-R1|32|8|256|4|0.582|0.303|
> |Qwen-2.5-7B-R1|28|4|112|8|0.775|0.193|
> |Qwen-3-4B-Thinking|36|8|288|4|0.565|0.369|
>
> To quantify this, we define a coarse heuristic "learned reasoning-head capacity" as:
> `score = num_layers * num_kv_heads * kv_group_size * mean`. The resulting values (595.97, 694.40, and 650.88) are of similar scale, indicating that RLKV discovers roughly comparable reasoning capacity across models. The less sparse distribution in Qwen-2.5-7B-R1 is therefore a consequence of its architecture, not a conflict with the sparsity objective.
>
> ---
>
> **W3: Feasibility of Jointly Training Gates and Model Parameters**
>
> We agree that jointly training the gates and model parameters is **feasible** and could potentially yield even better compression–performance trade-offs.
>
> Native Sparse Attention (NSA) [1] demonstrates this possibility: with large-scale pre-training, a model can learn to dynamically select important KV cache for sparse attention and achieve strong performance and efficiency. However, NSA relies on massive pre-training data and computing.
>
> In contrast, RLKV freezes the model parameters, which significantly reduces the optimization space and enables effective learning of head-importance distributions using only 3k examples and ~10 hours on 2×A100 GPUs.
> Performing such joint training in the post-training stage would face substantial challenges in training efficiency, including data requirements and computational cost. Moreover, jointly training the model parameters may lead to over-fitting to the training domain and hurt generalization.
>
> We therefore view joint training of gates and model parameters as an important but practically challenging direction for future work.
>
> [1] Native Sparse Attention: Hardware-Aligned and Natively Trainable Sparse Attention (2025)
>
> ---
> We hope our response addresses your concerns.

---

### Author Response · Authors · 2025-11-25
**Summary of New Experiments and Manuscript Revisions**

We sincerely thank the reviewers for their thoughtful feedback, constructive comments, and valuable insights. In this response, we provide new experimental results and note changes made in the revised manuscript.

---
**1. New Results on the Qwen-3-4B-Thinking**
We conduct main experiments on Qwen-3-4B-Thinking, as suggested by Reviewer `HVGo`, to further validate the effectiveness of RLKV on an additional model. The details are shown in Appendix Section A.5.

---
**2. New Results on the Four Subsets of MMLU-Pro**
We conduct additional experiments on four subsets of the challenging knowledge QA benchmark suite MMLU-Pro: Chemistry, Computer Science, Law, and Physics, as suggested by Reviewer `HVGo`, `Y94n`, and `Sm12`, to further validate the generalization ability of RLKV beyond the training math domain. The details are shown in Appendix Section A.6.

---
**3. Head Importance Analyses on Base Models**

We conduct head importance analyses on the base models Llama-3.1-8B-R1, Qwen-2.5-7B-R1, and Qwen-3-4B-Base, as suggested by Reviewer `vzQQ`. We observe similar performance degradation trends as the original head importance analyses on the reasoning models. These findings demonstrate that those heads identified by RLKV are important for the base models as well, suggesting that they rely on the full KV cache formed during the pre-training stage.

---
**4. Comparison Dynamic vs. Fixed Budget of R-KV and H2O**

We further explain why we use dynamic budgets instead of fixed budgets to evaluate R-KV and H2O. We add a relevant discussion in Appendix Section A.7, explaining the reasons why using a fixed budget in inference tasks may lead to unfair comparisons. In short, when the response length is much greater than the prompt length, using a global fixed budget makes the fixed budget unfair at the sample level.

Additionally, in Appendix Section A.8, we compare the performance of R-KV and H2O under dynamic and fixed budgets, as suggested by reviewer `Y94n`. The results show that R-KV and H2O perform worse under a fixed budget compared to a dynamic budget, which further supports the rationale for using a dynamic budget in our evaluation.

---
**5. Latency and Throughput Analyses**

We conduct latency and throughput analyses of RLKV, as suggested by Reviewer `Sm12`. We add a relevant discussion in Appendix Section A.9, presenting the results of latency and throughput of attention forward under different sequence lengths and sparsity levels, and end to end speedup results. The results of our current PyTorch implementation with Flash Attention without any optimizations achieve 9-16% end-to-end speedups on Llama-3.1-8B-R1 with Math500.ß We expect that integrating with modern inference frameworks could further improve speed.

---
**6. New Results of Layer-wise RLKV**

We conduct additional experiments on layer-wise RLKV, as suggested by Reviewer `Sm12`.
The results on Llama-3.1-8B-R1 with Math500 show that layer-wise RLKV significantly degrades performance compared to the original head-wise RLKV, further validating the inherent head distribution.

---
**7. Manuscript Revisions**

We have revised the manuscript to incorporate the above new results and clarifications, as suggested by the reviewers. Specifically:

- At line 39 of Section 1 (Introduction, Paragraph 1), we clarify that the KV cache size grows linearly with the context length, as suggested by Reviewer `Y94n`.
- In Section 1 (Introduction, Paragraph 2 and 3), we revise Figure 1 and the main motivation to avoid using a specific implementation to indicate general methodologies, as suggested by Reviewer `HVGo`. We add a sub-figure to illustrate general methodologies in Figure 1, and use "case study" to describe our specific implementation.
- At line 81 of Section 1 (Introduction, Paragraph 3), we use the detailed description to replace the undefined term "useless steps", as suggested by Reviewer `HVGo`.
- In Section 1 (Introduction, Paragraph 5), we reorganize our contributions to avoid over-claiming, as suggested by Reviewer `HVGo`.
- We moved the “Related Work” from Section 2 to Section 4 to avoid potential misunderstandings or unintended conceptual leaps. The new outline is: 1 Introduction, 2 Methodology, 3 Experiments, 4 Related Work, 5 Conclusion, and 6 Future Work.
- In Section 2 (Methodology, Paragraph 1), we reinstated the operational definition of “reasoning head” to clarify terminology and avoid potential misunderstandings.
- In Appendix Section A.1, we fix a typo of "AIME24", as suggested by Reviewer `Y94n`.
- In Appendix Section A.2, we add the training cost of RLKV, as suggested by Reviewer `Y94n`.
- In Appendix Section A.2, we add the prompt template, as suggested by Reviewer `Sm12`.

---
We sincerely thank the reviewers again for their efforts and valuable suggestions. We believe the new results and manuscript revisions have significantly improved the quality of our work. We look forward to any further feedback.

---

> ### Author Response · Authors · 2025-11-25
> **Details of New Results on the Qwen-3-4B-Thinking and New Results on the Four Subsets of MMLU-Pro**
>
> **New Results on the Qwen-3-4B-Thinking (Appendix Section A.5)**
>
> We conduct main experiments on Qwen-3-4B-Thinking. The details are shown in Appendix Section A.5.
> The following table reports the average accuracy across GSM8K, Math500, AIME24, and MBPP for 0.2, 0.4, 0.6, and 0.8 sparsity levels.
>
> | Qwen-3-4B-Thinking | full  | 0.2       | 0.4       | 0.6       | 0.8       |
> | ------------------ | ----- | --------- | --------- | --------- | --------- |
> | RLKV               | 0.743 | **0.779** | **0.758** | **0.592** | 0.070     |
> | DuoAttn            | 0.743 | 0.732     | 0.700     | 0.466     | **0.124** |
> | R-KV               | 0.743 | 0.693     | 0.540     | 0.310     | 0.109     |
> | H2O                | 0.743 | 0.255     | 0.083     | 0.014     | 0.009     |
>
> As shown, RLKV consistently outperforms all baselines at sparsity levels 0.2–0.6, achieving 5% higher accuracy than DuoAttention at 0.2 and 0.4 sparsity and 13% higher at 0.6 sparsity, demonstrating its effectiveness on this additional model.
>
> The following table reports the accuracy on each dataset at near lossless performance of RLKV.
>
> | Qwen-3-4B-Thinking | GSM8K (0.5) | Math500(0.5) | AIME24 (0.5) | MBPP (0.3)  |
> | ------------------ | ----------- | ------------ | ------------ | ----------- |
> | Full               | 0.951       | 0.776        | 0.433        | 0.812       |
> | RLKV               | **0.939**   | **0.828**    | **0.433**    | **0.824**   |
> | DuoAttn            | 0.930       | 0.724        | 0.200        | 0.802       |
> | R-KV               | 0.758       | 0.516        | 0.133        | 0.580       |
> | H2O                | 0.095       | 0.054        | 0.000        | 0.004       |
>
> As shown, at near lossless compression, RLKV matches or outperforms the full model on all four datasets.
>
> ---
> **New Results on the Four Subsets of MMLU-Pro (Appendix Section A.6)**
>
> We conduct additional experiments on four subsets of the challenging knowledge QA benchmark suite MMLU-Pro: Chemistry, Computer Science, Law, and Physics. Due to time constraints, we randomly sample 200 examples from each subset for evaluation on Llama-3.1-8B-R1 and Qwen-2.5-7B-R1. The details are shown in Appendix Section A.6.
>
> The following table reports the average accuracy across the four subsets for different sparsity levels:
>
> | Llama-3.1-8B-R1 | full  | 0.2       | 0.4       | 0.6       | 0.8       |
> | --------------- | ----- | --------- | --------- | --------- | --------- |
> | RLKV            | 0.436 | **0.421** | **0.411** | **0.371** | 0.036     |
> | DuoAttn         | 0.436 | **0.421** | 0.384     | 0.278     | **0.050** |
> | R-KV            | 0.436 | 0.384     | 0.325     | 0.242     | 0.044     |
> | H2O             | 0.436 | 0.312     | 0.155     | 0.040     | 0.000     |
>
> | Qwen-2.5-7B-R1 | full  | 0.2       | 0.4       | 0.6       | 0.8       |
> | -------------- | ----- | --------- | --------- | --------- | --------- |
> | RLKV           | 0.458 | **0.461** | **0.434** | **0.329** | 0.015     |
> | DuoAttn        | 0.458 | 0.414     | 0.386     | 0.226     | **0.029** |
> | R-KV           | 0.458 | 0.385     | 0.315     | 0.191     | 0.010     |
> | H2O            | 0.458 | 0.239     | 0.135     | 0.049     | 0.011     |
>
> As shown, RLKV demonstrates strong generalization beyond the math domain: it matches or outperforms the baselines at sparsity levels 0.2–0.6.

---

### Author Response · Authors · 2025-11-30
**General Response to Reviewers and ACs**

**To ACs:**

Thanks for your service to ICLR26! We respectfully submit this brief summary to assist you in evaluating our paper, which may facilitate a clearer understanding of the reviewing status and our rebuttal efforts.

---
**Paper Summary**

In this paper, we aim to compress the KV cache of reasoning LLMs to reduce memory usage and improve the throughput for efficient reasoning.

After analyzing existing KV compression methods why failing, we find that reasoning behaviors naturally require a full KV cache to maintain CoT consistency and they do not consider capturing dynamic reasoning behaviors that emerge during extended CoT sequences.
Therefore, we choose the head-reallocation paradigm to implement KV cache compression: keeping a part of heads with intact KV cache while others work with compressed constant KV cache.

Then, we propose RLKV, a novel reasoning-critical head identification method, which uses reinforcement learning to directly optimize the relationship between each head’s cache usage and reasoning quality.
Finally, using RLKV to guide KV cache compression consistently outperforms strong baselines, as shown in our extensive experiments.
Additionally, to our knowledge, RLKV is the first to identify a set of heads that matter for reasoning behaviors.

---
**Reviews Summary**

Before rebuttal, we received scores **8 / 4 / 4 / 2** with confidence **5 / 3 / 4 / 4**. Unfortunately, we did not receive any feedback after posting our rebuttals on Nov. 23.

**Strengths:**

Reviewers generally agree that:

1. Our problem formulation and motivation are interesting, insightful, right, fresh, and timely. It's also well supported by empirical evidence.
2. Our method is novel and efficient, which optimizes for semantic correctness rather than syntactic similarity, and learns a small number of extra gating parameters on frozen LLMs.
3. The performance of the proposed method is impressive.

**Weaknesses and Responses:**

We summarize the main concerns raised by reviewers and our corresponding responses below:

1. Response to Reviewer HVGo (Rating: 2)
    The main concerns are 3 points:
    - Writing clarity.
    We revise the manuscript in Figure 1 and the main motivation as suggested.
    - Definition of "reasoning heads".
    We note that the definition existed in the original version and reinstate it in Paragraph 2 of Section 2 Methodology in the revision.
    - Lack of novelty.
    We re-clarify our main innovations compared to DuoAttention in our response to HVGo, including a) the new problem formulation, and b) the RL-based optimization objective. We also provided clear evidence for the functional role of reasoning heads, compared to the DuoAttention in the original version.
    We revise our contributions to avoid over-claiming.
2. Response to Reviewer Sm12 (Rating: 4)
    The main concerns are 2 points:
    - Stability of head identification.
    We explain that we use the simplest prompt style without any tricks. We use the generalization experiments to further validate the stability.
    - Lack of end-to-end acceleration results.
    We add new experiments to report the latency and throughput of attention forwarding, and end-to-end speedup results with our current PyTorch implementation.
3. Response to Reviewer Y94n (Rating: 4)
    The main concerns are 3 points:
    - Generalization beyond the math domain.
    We conduct new experiments on four subsets of MMLU-Pro to validate the generalization ability of RLKV beyond the training math domain, further demonstrating the effectiveness.
    - Training cost vs. inference efficiency.
    We report the training cost and clarify the lightweight RL design and the inference-centric efficiency of RLKV.
    - Implicitly diminishing the baselines of H2O and R-KV with dynamic budgets.
    We further explain why we use dynamic budgets instead of fixed budgets to evaluate R-KV and H2O, and add new experiments showing that they with dynamic budgets outperform the fixed budgets.
4. Response to Reviewer vzQQ (Rating: 8)
    Compared to the other reviewers, he/she has minor concerns about reasoning head distribution and future directions. We conduct related new experiments and address detailed discussions.

---
**Conclusion**

We thank all the reviewers and ACs for improving our paper. The comments have been well taken and reflected in our revised paper.
In addition to the concerns, we are also encouraged that reviewers have recognized this work:

- Addressing an interesting, timely, and important problem.
- Showing strong empirical support and practical implications.
- Providing novel insights into reasoning behaviors through the lens of KV cache access patterns.

We believe that the extensive new experiments and detailed clarifications have effectively addressed the concerns raised by the reviewers.

Thanks for your precious time, ACs!

Best,
Authors

---

### Meta-Review · Area_Chair_cqa5 · 2026-01-08

**Summary:**

This paper argues that only a small subset of attention heads is truly critical for reasoning in large language models, and introduces an RL-based method to identify and preserve these heads while aggressively compressing others, achieving substantial KV cache savings with minimal impact on reasoning performance. From my reading of this paper and the reviews, this paper, although interesting, is still imperfect on methodological rigor and empirical support, as the concept of reasoning heads is weakly justified, baseline comparisons are potentially unfair, and key analyses on head roles, stability, and end-to-end speedups are missing. It also falls short on practical relevance, given the absence of training and total-cost analysis for the expensive RL phase, questionable generalization claims, and several technical inaccuracies. The authors tried to provide justification in the rebuttal phase, but, as they commented, there is no further feedback from the reviewers, which is a pity. By overall consideration of the review and the paper content, I tend to reject this paper.

**Reviewer Concerns:**

I think the presentation and experiment parts are improved based on the rebuttal information.

**Reviewer Scores:**

None of the reviewers replied in the rebuttal phase.

---

### Decision · Program_Chairs · 2026-01-26

Reject